# TMGBench: A Systematic Game Benchmark for Evaluating Strategic Reasoning Abilities of LLMs

## Abstract

The rapid advancement of large language models (LLMs) has accelerated their application in reasoning, with strategic reasoning drawing increasing attention. To evaluate the strategic reasoning capabilities of LLMs, game theory, with its concise structure, has become the preferred approach for many researchers. However, current research typically focuses on a limited selection of games, resulting in low coverage of game types. Additionally, classic game scenarios carry risks of data leakage, and the benchmarks used often lack extensibility, rendering them inadequate for evaluating state-of-the-art models. To address these challenges, we propose TMGBench, a benchmark characterized by comprehensive game type coverage, novel and diverse scenarios, and flexible game organization. Specifically, we incorporate all 144 game types summarized by the Robinson-Goforth topology of 2×2 games, which are constructed as classic games in our benchmark. Furthermore, we employ synthetic data generation techniques to create diverse, higher-quality game scenarios through topic guidance and human inspection for each classic game, which we refer to as story-based games. Lastly, to provide a sustainable evaluation framework adaptable to increasingly powerful LLMs, we treat the aforementioned games as atomic units and organize them into more complex forms through sequential, parallel, and nested structures. We conducted a comprehensive evaluation of mainstream LLMs, covering tests on rational reasoning, reasoning robustness, Theory-of-Mind capabilities, and reasoning in complex game forms. The results revealed that LLMs still have flaws in the accuracy and consistency of strategic reasoning processes, and their levels of mastery over Theory-of-Mind also vary. Additionally, o1-mini, the latest reasoning model from OpenAI, was also evaluated across the sequential, parallel, and nested game structures and reached accuracy rates of 66.6%, 60.0%, and 70.0%, respectively, highlighting the challenges posed by TMGBench.

## 1 Introduction

The rapid advancement of large language models (LLMs) has reshaped the paradigm of artificial intelligence, achieving breakthroughs across various domains (Zhao et al., 2023; Huang & Chang, 2022; Lewkowycz et al., 2022; Huang et al., 2022; Paranjape et al., 2023). These achievements are largely attributed to LLMs' ability to assimilate vast amounts of knowledge during training, emerging with the capacity to organize information at a coarse level and link knowledge at a fine-grained level through their internal representations (Min et al., 2023; Zhao et al., 2023). These core capabilities have driven the success of LLMs in numerous reasoning tasks, including mathematical reasoning (Hendrycks et al., 2021; Zhang et al., 2023), commonsense reasoning (Sap et al., 2019; Bisk et al., 2020), logical reasoning (Lei et al., 2023), and strategic reasoning (Lorè & Heydari, 2023; Duan et al., 2024). Among these, strategic reasoning has attracted considerable attention due to its multi-agent nature and close association with social intelligence (Gandhi et al., 2023).

Strategic reasoning refers to the cognitive process of anticipating, planning, and responding to others' actions to achieve specific objectives within competitive or cooperative contexts (Zhang et al., 2024a). Consequently, game scenarios—naturally involving both cooperation and competition—have intuitively become a fertile ground for studying LLMs' strategic reasoning abili-

ties (Brookins & DeBacker, 2023). In particular, researchers have engaged LLMs in game-playing, analyzing their decision-making behaviors and evaluating their strategic intelligence in such scenarios (Duan et al., 2024). The Prisoner's Dilemma, as one of the most classic game theory scenarios, has been extensively studied in this context (Herr et al., 2024). Additionally, other traditional games such as the Battle of the Sexes (Kreps, 1990), the Stag Hunt (Carlsson & Van Damme, 1993), and the Dictator Game (Forsythe et al., 1994) have also drawn significant attention. These studies provide initial insights into the strategic reasoning capabilities of LLMs (Horton, 2023; Brookins & DeBacker, 2023; Phelps & Russell, 2023; Akata et al., 2023; Li et al., 2023; Aher et al., 2022).

However, current research has three major limitations, hindering a comprehensive, robust, and sustainable evaluation of LLMs' strategic reasoning capabilities: (1) *Limited coverage of game types*: Most studies focus on a handful of classic games without considering the full diversity of game structures. (2) *Potential risk of game scenario leakage*: Classic game scenarios are likely to be present in the training corpus, raising concerns over data leakage. (3) *Poor extensibility of game forms*: Existing studies primarily focus on a narrow range of game forms, which may no longer suffice to challenge high-performing LLMs such as o1-mini from OpenAI.

To address the above issues, we introduce TMGBENCH, a benchmark that encompasses a comprehensive range of game types, features synthesized game scenarios, and supports scalable and reorganizable game forms. Specifically, *to address the first issue*, we include all 144 game types defined by the Robinson-Goforth topology of 2x2 games (Robinson & Goforth, 2005). This topology encompasses a variety of game structures based on different numerical payoff matrices, including but not limited to classic games like the Prisoner's Dilemma(§2.2). *To address the second issue*, we employ synthetic data generation techniques to create five different story-based games for each classic game. In essence, a story-based game is a contextual framing counterpart of its corresponding classic game, sharing the same structure but differing in context (Lorè & Heydari, 2023). To ensure high-quality data synthesis, we introduce two additional steps: topic control and human inspection. We first define a set of topics commonly associated with cooperation and competition, such as business and law, to guide the data generation process. Then, to ensure that the synthesized games meet the required game structures and are easily understandable, we conduct rigorous human inspection (§2.3). *To address the third issue*, we propose three forms for expanding and organizing games: sequential, parallel, and nested. Using the above constructed games as atomic units, we reorganize them into these complex forms to assess the strategic reasoning of LLMs. The sequential and parallel forms evaluate the model's capacity for sequential and parallel decision-making, respectively, while the nested form explores the LLMs' multi-layered strategic reasoning abilities (§2.4).

Based on TMGBENCH, we conduct comprehensive analyses and evaluations of current mainstream LLMs (§3), including assessments of rational reasoning, reasoning robustness, Theory-of-Mind (ToM) capabilities, and reasoning in complex game forms, leading to the following key findings:

(1) Advanced LLMs like gpt-4o demonstrate strong strategic reasoning, with over 80% accuracy, but struggle to generalize across contexts and scenarios. Models like claude-3-5-sonnet further reveal this inconsistency, with performance variability marked by coefficients of variation nearing 0.5.

(2) Though GPT models often perform well, their reasoning inconsistency on certain task sub-types is marked by an 'asymmetric pattern' as reported, which is the main cause of the statistical biases.

(3) Several top-tier LLMs demonstrate stable first-order ToM abilities, with some effectively utilizing second-order ToM for comparable tasks. In contrast, models such as Llama-3.1-70B appear restricted to first-order reasoning.

(4) Complex-form games that are derived from atomic units in TMGBENCH present considerable challenges for LLMs, including those with strong reasoning abilities like o1-mini from OpenAI, which often struggle as the number of games increases.

## 2 TMGBENCH

### 2.1 BENCHMARK OVERVIEW

TMGBENCH is a benchmark designed to evaluate the strategic reasoning capabilities of LLMs in game-theoretic scenarios, illustrated by Figure 1. It comprehensively covers 144 types of games

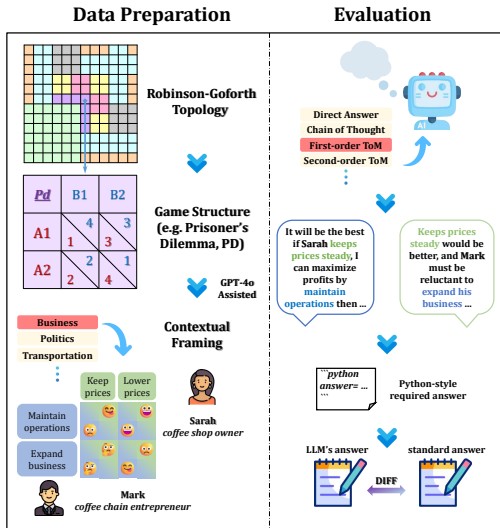

Figure 1: An concept map of TMGBENCH. The data preparation of the benchmark includes 3 ingredients: Robinson-Goforth topology, game structure and contextual framing. The evaluation of the benchmark embraces several prompting methods (including ToM promptings) to elicit strategic reasoning process of LLMs.

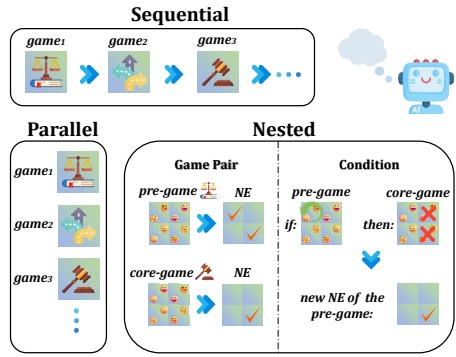

Figure 2: We design several complex forms of strategic reasoning tasks using TMGBENCH. which include: (1) sequential form, where LLMs are required to respond to multiple game tasks in a row, with history of previous tasks; (2) parallel form, where LLMs are required to response multiple game tasks simultaneously; (3) nested form, where LLMs are required to response a set of interlinked game tasks (in our settings, we relate to them as pre-game and core-game). Games in the complex forms can be selected with different game structures and various contexts.

(see §2.2), with each type containing multiple instances (in each instance, there are two players and each player can choose between two strategies, resulting in four possible situations), which can be categorized into classic and story-based settings. Notably, the story-based instances are produced using synthetic data generation techniques and are grounded in real-life themes, effectively mitigating the issue of data leakage (see §2.3). Furthermore, each game in TMGBENCH can be treated as an atomic unit, and multiple atomic games can be structured in a more complex task with parallel, sequential, or nested form (see §2.4). These complex scenarios effectively facilitate the evaluation of advanced LLMs' abilities in parallel, sequential, and multi-layered decision-making. To precisely evaluate the reasoning abilities of LLMs, we use their performance in inferring the optimal strategy combination, i.e., the Nash equilibrium, as the evaluation criterion. Additionally, the designed evaluation metrics provide a fine-grained assessment of the robustness and self-consistency of LLMs' strategic reasoning abilities (see §2.5).

## 2.2 GAME TOPOLOGY

Although previous research has explored LLMs' reasoning abilities within the context of game theory, existing studies have primarily focused on a few well-known games, such as the Prisoner's Dilemma, Battle of the Sexes, and Stag Hunt (Brookins & DeBacker, 2023; Phelps & Russell, 2023; Guo, 2023). However, these studies cover a limited game types, resulting in incomplete evaluations. Thereby, a broader variety of games is urgently needed to conduct a *systematic* assessment of LLMs.

To address this, we incorporate 144 game types (we later refer to a type as an equivalence class) based on the Robinson-Goforth topology of 2×2 games (Robinson & Goforth, 2005). Classic games like the Prisoner's Dilemma belong to one of the equivalence classes within this topology. Specifically, the topology of 2×2 games elegantly illustrates the relationships among strictly ordinal 2×2 games, each with a unique payoff structure, leading to different dominant strategies, Nash equilibria, and reasoning approaches (more details in Appendix C.1). We categorize all the 144 games with numerical payoffs from the original topology into the *classic setting* tasks. Due to space constraints, we provide an introduction to the Robinson-Goforth topology in Appendix C.2.

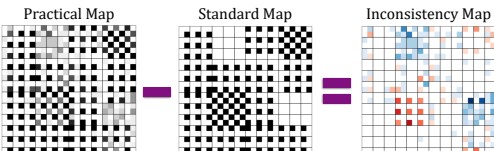
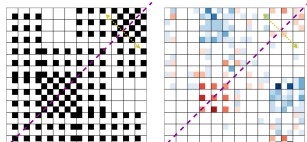

Figure 3: Demonstration of the inconsistency heat map. Each of the grids is divided into 4 quarter-grids, indicating the 4 situations. By subtracting the standard map from the practical map element-wise, we get the inconsistency map, where blue colours indicate positive difference and red colours indicate negative difference. The deeper the colour means the larger the difference between the LLM's response and the standard answer.

Figure 4: Axisymmetry in heat maps can be illustrated by the left sub-figure, where the standard heat map exhibits perfect axisymmetry across the counter-diagonal. In contrast, LLMs' responses tend to demonstrate quasi-axisymmetry, as shown by the right sub-figure. Certain pairs of positions fail to align precisely when reflected across the axis and may exhibit discrepancies, deviating from the ideal symmetric pattern.

## 2.3 CONTEXTUAL FRAMING

Relying on the Robinson-Goforth topology, we can systematically construct all types of classic setting tasks. However, this alone is insufficient, as games often take place in diverse real-life contexts, involving different topics, types of participants and their preferences. Such contextual framing of games introduces new challenges for LLMs (Lorè & Heydari, 2023).

To further explore LLMs' strategic reasoning capabilities in real-world scenarios, we use classic games as seed data and employ synthetic data generation techniques, leveraging GPT-4o to construct story-based games. Specifically, in story-based games, we replace the pure game information of classic games with real-life scenarios, covering topics such as business, law and transportation. Additionally, the two players are substituted with characters representing broader semantics (e.g., people, animals, organizations, and even nations), and the payoff values are transformed from pure numbers into specific states or rewards relevant to the characters. For each classic game, we generate 5 corresponding story-based games.

To ensure high-quality data generation, we undertake the following steps: First, we use GPT-4o to synthesize the contextual data. Second, we design precise prompts to ensure the generated data adhere to the given game structures. Third, we select topics from real-life scenarios where strategic interactions are common, guiding the data generation process. Finally, we conduct rigorous human reviews to ensure the data's quality and diversity.

Details on the data generation process, prompts, human review procedures, and topic distribution of the data can be found in Appendix D.

## 2.4 COMPLEX FORMS

The 2×2 games in the topology represent a highly condensed game structure. However, in real life, we often encounter more complex game forms, such as making continuous decisions, making multiple decisions simultaneously, or considering the impacts of one decision on another.

To evaluate LLMs' strategic reasoning abilities with more constraints, we treat the aforementioned individual games as atomic games and expand them in three forms: sequential, parallel, and nested. The organization of these forms is illustrated in Figure 2. Specifically, in the *sequential form*, we randomly sample multiple games from the story-based games, requiring the LLM to make decisions sequentially. Only if the LLM provides correct answers for all games is it considered to have made correct decisions. In the *parallel form*, the LLM is given multiple randomly sampled games and must make decisions simultaneously. Similarly, the LLM is deemed to have made correct decisions only if it solves all games correctly. In the *nested form*, we randomly sample two games, designated as the `pre-game` and the `core-game`, where the `core-game` holds greater importance. The decisions made by the LLM in the `pre-game` affect the strategy space in the `core-game`. Thus, the LLM is judged to have made correct decisions only if it demonstrates forward-looking

reasoning by choosing a sub-optimal solution in the `pre-game` to achieve the optimal solution in the `core-game`. We demonstrate a template to generate an nested form game in Appendix F.2.

Theoretically, using these atomic games, we can expand the framework to generate infinitely many increasingly complex game forms, thereby providing a continuous benchmark for evaluating the performance of more advanced LLMs.

## 2.5 EVALUATION METRICS

As explained in Section 2.2, our benchmark is perfectly suitable to display in a 12x12 square table, each grid representing one of the 144 equivalence classes. In the evaluation process we conduct *repetitive* tests in every data point of each equivalence class. Each test starts with the input of the setting (classic/story-based) and the question, and ends with LLM's response containing a list of choices corresponding to multiple choices or no choice (when the given list is empty).

**Notation**. For notation, we assign $\text{Freq}_{i,j,o}$ as the frequency of the $o$-th choice happening to be in the tests of the grid at $i$-th row, $j$-th column, where the 1, 2, 3 and 4-th choice correspond to the upper-left, upper-right, lower-left and lower-right quarter-grid respectively.

**Inconsistency Heat Map**. According to conclusions of the Robinson-Goforth topology (Robinson & Goforth, 2005), we convert the standard answer of each equivalence class into a heat map named the *standard heat map*, with the coloured quarter-grid to be the choice in the standard answer. Similarly, as for practical result provided by LLMs, we set the value of $\text{Freq}_{i,j,o}$ as the colour depth of each quarter grid, which builds up the *practical heat map*. Naturally, we subtract the standard heat map from the practical heat map in an element-wise manner to get the *inconsistency heat map*, which is a standardised tool for our evaluation, shown in Figure 3.

**Inconsistency Degree**. In order to display the quantified performance of LLMs, we extract inconsistency degree from a map, which helps reveal the gap between LLMs' response and standard answer, and it is defined as

$$\text{ID} = \frac{1}{144} \sum_{i=1}^{12} \sum_{j=1}^{12} \frac{1}{4} \sum_{o=1}^{4} \Delta\text{Freq}_{i,j,o}^2$$

where $\Delta\text{Freq}_{i,j,o}$ indicates the the difference (between the LLM's answer and the standard answer) of frequency of the $o$-th choice at $i$-th row, $j$-th column.

**Bias Degree**. Owing to the symmetric property of the topology framework of 2×2 matrix games, the distribution of answers over the heat map has axial symmetry by the counter-diagonal (Figure 4). Motivated by this elegant property, we set up another metric to evaluate the bias degree of LLMs' answers, which we expect robuster LLMs to display lower degrees of bias. The bias degree reflects the stability and symmetry of LLMs' strategy, and it is defined as

$$\text{BD} = \frac{1}{144} \sum_{i=1}^{12} \sum_{j=1}^{12} \frac{1}{4} \sum_{o=1}^{4} (\text{Freq}_{i,j,o} - \text{Freq}_{j,i,\text{ref}_o})^2$$

where the meaning of $\text{ref}_o$ is the index of choice $o$'s counterpart considering the reflection operation by the counter-diagonal, and we have the mapping relation: $\{1, 2, 3, 4\} \mapsto \{4, 2, 3, 1\}$. (e.g. $\text{ref}_1 = 4$ means that the reflection counterpart of choice 1 is choice 4, vice versa)

**Perfect Accuracy Rate**. In addition to the metrics mentioned above, we also set up a more rigorous metric named perfect accuracy rate, which ignores the partially correct answer and only considers perfectly correct answer in each test, and it is defined as

$$\text{PAR} = \frac{1}{144} \sum_{i=1}^{12} \sum_{j=1}^{12} \frac{1}{T} \sum_{t=1}^{T} \mathbb{I}\{\text{rsp}_{t,i,j} = \text{std}_{i,j}\}$$

which means that we count only if the response perfectly matches the standard answer, where $T$ represents the number of times we invoke a LLM to response on a certain game task.

**Metrics with Subscript**. As a matter of fact, within the topology, different equivalence classes have different number of Nash equilibria (ranging from $\{0, 1, 2\}$), leading to a discrepancy in reasoning

Table 1: Overall statistics of LLMs' performance on classic setting tasks. The up arrow(↑) means the larger value indicates better performance, while the down arrow(↓) means the smaller value indicates better performance. All values are expressed as percentages.

| Family | Model | Metric / Prompting | | | | | |
| | | PAR(↑) | | ID(↓) | | BD(↓) | |
| | | DA | CoT | DA | CoT | DA | CoT |
|---|---|---|---|---|---|---|---|
| GPT | gpt-4o | 52.08 | **80.38** | 16.81 | **3.78** | 28.49 | **7.79** |
| | gpt-4o-mini | 14.93 | 74.02 | 27.15 | 4.38 | 48.59 | 8.29 |
| | gpt-3.5-turbo | 30.21 | 34.38 | 27.64 | 17.87 | 50.15 | 30.19 |
| Claude | claude-3-5-sonnet | **59.38** | 79.69 | **14.79** | 7.13 | **27.76** | 14.34 |
| | claude-3-haiku | 24.31 | 40.28 | 39.58 | 25.17 | 72.22 | 44.10 |
| Llama | Llama-3.1-70B | 13.02 | 54.29 | 36.15 | 15.32 | 40.71 | 26.63 |
| | Llama-3.1-8B | 18.75 | 22.63 | 38.49 | 31.19 | 81.32 | 47.64 |
| Qwen | Qwen2-72B | 43.06 | 46.21 | 26.30 | 19.94 | 35.59 | 29.29 |

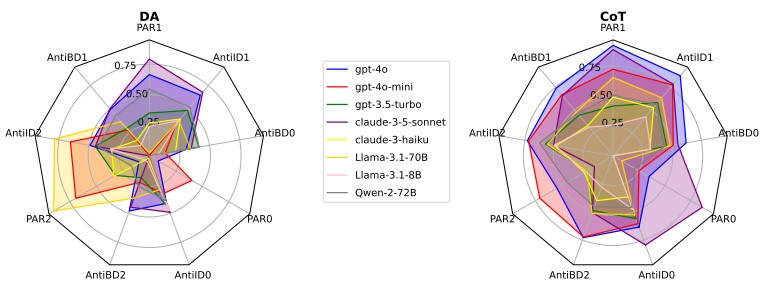

Figure 5: Radar charts of the 9 sub-metrics of 8 LLMs' performance, comparing the DA prompting (left side) and the CoT prompting (right side). AntiID and AntiBD are derived from ID and BD while higher values indicate better performances (in order to consistent with PAR).[1]

difficulty, therefore we propose metrics with subscript that represents for different types of equivalence groups (we refer them to 0-task, 1-task, 2-task respectively), which we refer to as sub-metrics. Therefore we have $ID_n, BD_n, PAR_n (n = 0, 1, 2)$ which means the inconsistency degree, the bias degree, and the perfect accuracy rate across all equivalence classes that have $n$ equilibra.

## 3 ANALYSIS

### 3.1 OVERVIEW OF LLMS' PERFORMANCE

Overall, we select several SOTA models according to Open LLM Leaderboard (Fourrier et al., 2024) and conduct extensive experiments on TMGBENCH. These models include GPT (gpt-4o-2024-05-13, gpt-4o-mini-2024-07-18, gpt-3.5-turbo-0125), Claude (claude-3-5-sonnet-20240620, claude-3-haiku-20240307), Llama (Llama-3.1-8B, Llama-3.1-70B), and Qwen (Qwen2-72B). We perform 4 independent tests on each data point, covering both the classic setting and the story-based setting (thus we conduct 2,880 tests to generally evaluate a certain model). During the evaluation, we set the temperature of the tested LLMs to 0 or near 0, ensuring the lowest degree of uncertainty and enhancing the faithfulness of our evaluation. More details of the evaluation process are provided in Appendix D.1.

**Games in TMGBENCH are not easy for most LLMs.** First we overall evaluate how well LLMs can behave on the classic setting tasks of our benchmark, to assess their basic capability of strategic

[1] $AntiBD = 1 - \sqrt{BD}$, $AntiID = 1 - \sqrt{ID}$

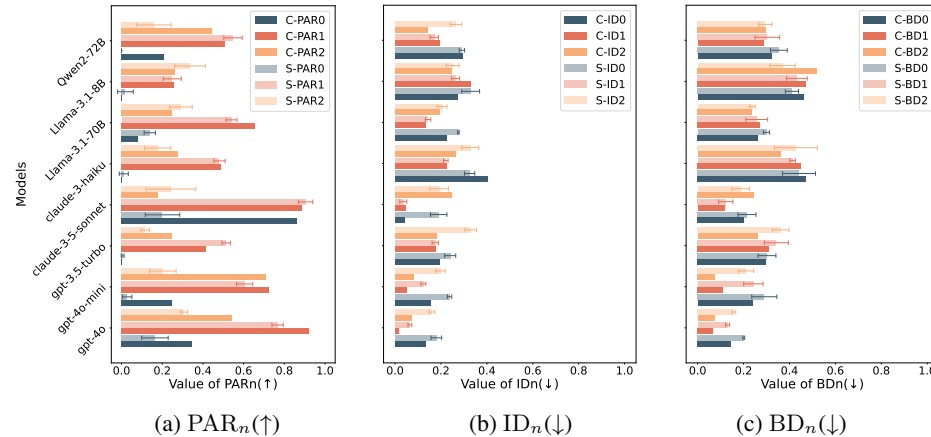

(a) $\text{PAR}_n(\uparrow)$     (b) $\text{ID}_n(\downarrow)$     (c) $\text{BD}_n(\downarrow)$

Figure 6: Comparison of LLMs' performance under the classic setting (indicated by '$C$-' label, in opaque colour) and the story-based setting (indicated by '$S$-' label, in semi-opaque colour with error bar), where the length of the bars represent the value of metrics, and the error bars represent the standard deviation over all 5 data points of the story-based setting tasks.

reasoning. We initially adopt two basic prompting methods: Direct Answer (DA) prompting and Chain-of-Thought (CoT, (Wei et al., 2022)) prompting, which represent shallower, faster thinking patterns and deeper, slower thinking patterns, respectively.

As seen from Table 1, gpt-4o, gpt-4o-mini and claude-3-5-sonnet are more capable compared to other models, with a high overall accuracy rate (around 80%) and low inconsistency and low bias score (around 5%). Specifically, as shown in Figure 5 formed by 9 sub-metrics, gpt-4o performs the best on 1-tasks, gpt-4o-mini beats others on 2-tasks, and claude-3-5-sonnet are relately better at 0-tasks. Moreover, comparing the performance of employing DA prompting and CoT prompting, we find that CoT prompting almost provides comprehensive improvement but few exceptions like the $\text{PAR}_2$ of Llama-3.1-70B.

Despite the excellent performance of the top-tier models (gpt-4o and claude-3-5-sonnet), other models often do not exhibit robust performance across all 3 different types of tasks. The inconsistency degree and bias degree in these models can be more than double or triple those of the top-performing models. This indicates that from a systematic point of view, even classic setting tasks from TMG-BENCH are challenging for most LLMs.

**LLMs' performance is vulnerable across various narratives.** At the theoretical level, we consider classic setting tasks and story-based tasks to be fundamentally the same problems within the domain of game theory. However, this conclusion appears not transferable to LLMs at the practical level. For LLMs, the complexity and nuance of story-based tasks introduce unique challenges, where LLMs are required to be robust in understanding and reasoning concurrently.

In Figure 6, we compare the performance of LLMs using CoT prompting, which is more robust according to previous analysis. The figure reveals the vulnerable performance of LLMs on tasks in story-based setting (corresponding to various narratives), marked by two primary characteristics:

(1) The advanced models, specifically gpt-4o, gpt-4o-mini and claude-3-5-sonnet, exhibit significant performance degradation. Notably, gpt-4o demonstrates a broad under-performance across the board, while gpt-4o-mini experiences the most pronounced decline in performance on 2-task scenarios, where its $S\text{-PAR}_2$ metric falls to less than one-third of its $C\text{-PAR}_2$ counterpart. Similarly, claude-3-5-sonnet shows the largest performance drop in 0-task, with its $S\text{-PAR}_0$ metric reduced to less than one-fourth of $C\text{-PAR}_0$, and its $S\text{-ID}_0$ metric exceeding four times that of $C\text{-ID}_0$.

(2) The performance of certain localities exhibits significant fluctuations. A particularly notable degradation occurs in the PAR scores for 0-task and 2-task scenarios handled by claude-3-5-sonnet, where the coefficients of variation $c_v$ (defined as $c_v = \frac{\sigma}{\mu}$, with $\sigma$ representing the standard deviation and $\mu$ the mean) approach 0.5. These eminent values of $c_v$ suggest a lack of robustness in performance across different narratives.

Table 2: Performance of LLMs using different ToM compared to CoT. Text in red color indicates the performance gets better and text in blue color indicates the performance gets worse (both compared to CoT). Bold text means the best performance across the three prompting methods. Grey areas mean an LLM is good at using some kind(s) of ToM. All values are expressed as percentages.

| Model | Prompting | 0-Task | | | 1-Task | | | 2-Task | | |
|---|---|---|---|---|---|---|---|---|---|---|
| | | $PAR_0(\uparrow)$ | $ID_0(\downarrow)$ | $BD_0(\downarrow)$ | $PAR_1(\uparrow)$ | $ID_1(\downarrow)$ | $BD_1(\downarrow)$ | $PAR_2(\uparrow)$ | $ID_2(\downarrow)$ | $BD_2(\downarrow)$ |
| gpt-4o | CoT | 34.72 | 13.37 | 14.41 | 92.36 | 1.58 | 6.76 | **54.17** | **7.38** | **7.38** |
| | FoToM | **43.06** | **9.46** | **9.81** | **95.14** | **0.72** | **4.14** | 50.00 | 8.94 | 8.59 |
| | SoToM | 31.94 | 9.81 | 10.68 | 91.67 | 1.45 | 6.00 | 52.78 | 7.99 | 8.16 |
| gpt-4o-mini | CoT | **25.00** | **15.62** | 23.94 | 72.45 | 5.08 | 11.09 | **70.83** | 7.97 | 7.69 |
| | FoToM | **25.00** | 19.53 | **19.53** | **99.54** | **0.03** | **5.08** | 47.22 | 10.59 | 10.59 |
| | SoToM | 18.06 | 26.56 | 26.22 | 98.84 | 0.19 | 5.38 | 68.06 | **5.38** | **5.38** |
| gpt-3.5-turbo | CoT | 0.00 | **19.44** | 29.69 | 41.67 | 17.55 | 30.95 | 25.00 | 18.23 | 26.13 |
| | FoToM | 0.00 | 21.44 | **22.83** | **54.40** | 19.30 | 42.52 | 0.00 | 37.85 | 59.20 |
| claude-3-5-sonnet | CoT | **86.11** | **4.25** | 20.23 | 88.89 | 4.72 | 11.68 | 18.06 | 24.48 | 24.48 |
| | FoToM | 68.06 | 7.73 | **16.06** | **92.13** | **2.56** | **7.74** | **47.22** | 15.10 | 15.10 |
| | SoToM | 47.22 | 21.35 | 28.99 | 90.05 | 4.05 | 14.38 | 33.33 | **14.93** | **14.93** |
| claude-3-haiku | CoT | 0.00 | 40.28 | 47.22 | **49.07** | 22.45 | 44.91 | **27.78** | 26.39 | 36.11 |
| | FoToM | 0.00 | **33.33** | **37.50** | 47.22 | **22.22** | 48.61 | 11.11 | 43.06 | 56.94 |
| Llama-3.1-70B | CoT | 8.33 | 22.47 | **26.43** | **65.59** | 13.43 | 27.16 | 25.00 | 19.53 | 23.70 |
| | FoToM | 2.78 | 30.82 | 35.59 | 49.54 | 18.68 | 27.49 | **69.44** | **6.08** | **22.74** |
| | SoToM | **23.61** | **21.27** | 28.73 | 60.42 | 14.09 | **23.70** | 12.50 | 24.05 | 25.26 |
| Llama-3.1-8B | CoT | 0.00 | 27.34 | **46.09** | 25.77 | 32.90 | 47.17 | 26.39 | 24.74 | 52.00 |
| | FoToM | 0.00 | **22.14** | 59.20 | **27.55** | **31.97** | 67.18 | 15.28 | 33.64 | 65.49 |
| Qwen2-72B | CoT | 20.83 | 29.25 | 32.20 | 50.78 | 19.35 | 28.73 | 44.44 | 14.15 | 29.77 |
| | FoToM | 0.00 | 36.46 | 35.07 | 45.14 | 26.92 | 49.54 | 11.11 | 37.50 | 49.13 |

## 3.2 FINDINGS OF LLMS' BEHAVIOURS

**LLMs demonstrate first/second-order ToM abilities.** In tasks across all equivalence classes, 1-tasks have the lowest reasoning difficulty because at least one player has a dominant strategy, which means the player can make an unconditionally optimal decision regardless of the counterpart's choice. In such cases, once a player (denoted as A) can make this unconditionally optimal decision, their counterpart (B) can, using first-order Theory-of-Mind (ToM), easily determine the best response for themselves (B).

This insight motivated us to apply FoToM prompting to LLMs, representing the **F**irst-**o**rder **T**heory-**o**f-**M**ind thinking, to aid in solving these tasks. As seen in Table 2, top-tier models like gpt-4o show improvement in both 0-tasks and 1-tasks when utilizing FoToM. Model claude-3-5-sonnet improves on 1-tasks and 2-tasks, and gpt-4o-mini displays a significant surge in performance on 1-tasks and so does Llama-3.1-70B on 2-tasks. However, for models like Llama-3.1-8B and Qwen2-72B, FoToM does not seem to provide any prominent advantage and may even result in worse performance. Notably, no LLM achieves overall improvement across all task categories by merely using first-order ToM, and 0-tasks appear to be the most challenging for LLMs to solve.

Furthermore, we wondered if LLMs display some ability to use first-order ToM could also be capable of second-order ToM. According to Liddle & Nettle (2006), higher-order ToMs are generally more difficult to master than first-order ToM. Thus we selected only advanced models that demonstrated proficiency in first-order ToM to attempt solving specific tasks using **S**econd-**o**rder **T**heory-**o**f-**M**ind (SoToM) prompting. As seen in Table 2, models like gpt-4o, gpt-4o-mini and claude-3-5-sonnet show consistent performance when applying second-order ToM to tasks they are already capable of solving better with first-order ToM. However, the improvements from using SoToM generally do not exceed those achieved with first-order ToM. In addition, Llama-3.1-70B's underperformance with SoToM suggests that possessing first-order ToM capabilities does not necessarily imply proficiency with second-order ToM. The prompts used for FoToM and SoToM are provided in Appendix D.2.

**Certain behavioural pattern contributes to poor performance.** Based on the analysis from the previous sections, it is encouraging to note that top-tier LLMs demonstrate high accuracy and low inconsistency when solving 1-task scenarios, regardless of the prompting used (CoT, FoToM, or SoToM). However, their performance declines significantly when addressing other types of tasks.

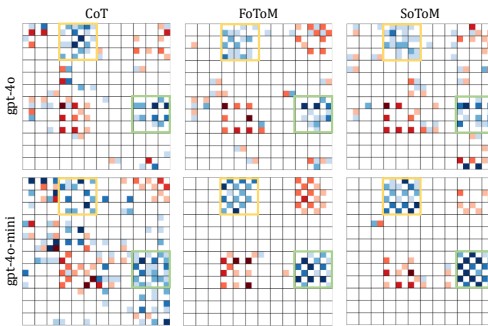 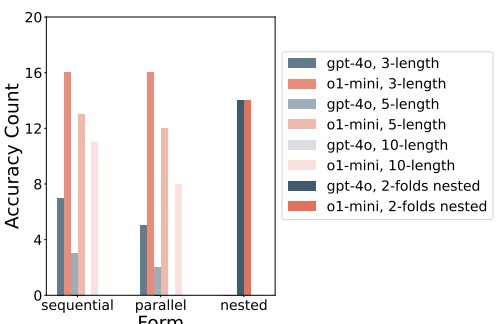

Figure 7: Inconsistency heat map of GPT series models using different prompting methods. The yellow boxes and green boxes represent the 0-task areas in the topological framework.

Figure 8: Top LLMs' performance on the games in complex forms of three types. Owing to the expensive inference cost, we run 20 times for each configuration.

For the advanced GPT series models, it is particularly noteworthy that they perform the worst on 0-tasks out of all types. Apart from the low PAR and high ID on 0-tasks compared to 1-tasks, the bias degree also doubles (for gpt-4o) or even several times higher (for gpt-4o-mini). Surprisingly, as illustrated in Figure 7, these models display a similar answering pattern that appears non-coincidental. Within the topological framework, there are two square areas representing 0-tasks (enclosed in yellow boxes and green boxes), which should theoretically be symmetric across the counter-diagonal. The standard heat map of these two areas is entirely blank, reflecting no existing equilibrium, so the two areas of the inconsistency heat maps just reflect the distribution of LLMs' practical responses.

Under closer inspection, it becomes evident that the models exhibit a consistent pattern when addressing 0-tasks. In yellow-box areas, their answers tend to emphasize the upper-right and lower-left quarter-grids, whereas in green-box areas, their answers tend to emphasize the upper-left and lower-right quarter-grids. This pattern appears to be the primary cause of the high bias degree. However, the phenomenon is quite counter-intuitive: it introduces a strong asymmetry along the counter-diagonal. In other words, simply swapping the id of two players and their actions, which does not alter the fundamental game structure, leads the LLMs to identify different Nash equilibria. Nevertheless, it is quite strange for them to provide such uniform "wrong answers" within each box, while the answers across the two boxes are entirely asymmetric.

To testify that this is not due to the position bias in the prompts (refer to the *FoToM* prompting and *SoToM* prompting in Appendix D.2), we design the *reFoToM* prompting and the *reSoToM* prompting (refer to the *reFoToM* prompting and *reSoToM* prompting in Appendix D.2) which swap the order of the players happens in the FoToM prompting and the SoToM prompting respectively. The results in Appendix E.1 imply that such 'asymmetric inconsistency pattern' is not strong related to the orders in the prompt. We demonstrate two typical examples of this phenomenon in Appendix E.2.

**Complex forms bring more challenging tasks.** To verify that TMGBENCH can be extended to harder tasks which may better align with complicated scenarios from the reality, we run the test on the three complex forms we mention in Section 2.4, to assess the performance of two strongest LLMs (o1-mini and gpt-4o) in complex strategic reasoning.

We setup the test by dividing it into several types: (1) in sequential form and parallel form, we set the variable of number of the games from the set $\{3, 5, 10\}$; (2) in nested form, we just use some 2-folds nested games (due to the high verification cost when the number increases).

As seen from Figure 8, the top-tier model gpt-4o has a dramatically low accuracy rate in either sequential or parallel games, even the strongest reasoning model o1-mini still failed at times; when the number of the games increase, their performances both drop, which is consistent with intuition. As for the games of nested form, two models' performances are relatively reasonable, while it is fair to infer that if we increase the number of layers of the games that in the nested structures, it will present a great challenge for LLMs. The overall accuracy rates of o1-mini over the three forms are 66.6%, 60.0% and 70.0% respectively, while gpt-4o performs worse, with accuracy rates reaching only 50.0%, 35.0% and 70.0% respectively.

## 4  RELATED WORK

**Strategical Reasoning of LLMs.** Large language models have made notable breakthroughs in reasoning tasks, such as mathematical, causal, and commonsense reasoning, enabling their increasing use in complex tasks that support human decision-making (Imani et al., 2023; Kıcıman et al., 2023; Zhao et al., 2024). This progress has sparked a growing interest in studying their strategic reasoning capabilities (Zhang et al., 2024a). Game theory, with its highly abstract representation of real-world strategic scenarios, has garnered significant attention from researchers (Duan et al., 2024; Huang et al., 2024). The prisoner's dilemma, as one of the most classical games, has been widely used to evaluate the strategic reasoning abilities of LLMs (Brookins & DeBacker, 2023; Guo, 2023; Akata et al., 2023; Phelps & Russell, 2023; Xu et al., 2023). In addition, several well-known game theory scenarios, such as the Dictator Game (Horton, 2023; Fan et al., 2023; Brookins & DeBacker, 2023), the Ultimatum Game (Aher et al., 2022), the Public Goods Game (Li et al., 2023) and the Battle of the Sexes (Akata et al., 2023), have been employed to evaluate LLMs' capabilities. However, current studies often focus on individual games, resulting in incomplete assessments and less robust conclusions. To address this, we propose TMGBENCH, a benchmark for evaluating LLMs by 2×2 games, where its atomic games can be further organized using sequential, parallel, and nested formats to provide an in-depth evaluation of the SOTA models gpt-4o and o1-mini.

**Theory-of-Mind of LLMs.** Theory-of-Mind (ToM) refers to the ability to understand and infer human mental states (Premack & Woodruff, 1978). Due to the multi-player nature of game theory, players' ability to reason about the "minds" of other participants is crucial. Existing research has initiated discussions on whether machines possess ToM capabilities. For instance, Kosinski (2023) suggested that ToM might emerge spontaneously in LLMs, as demonstrated through assessments using false-belief tasks. However, (Ullman, 2023) argued that such successes are fragile, easily disrupted by minor perturbations that would not affect an entity genuinely possessing ToM. Nevertheless, many researchers propose enhancing LLMs' strategic reasoning abilities by incorporating ToM. Guo et al. (2023) designed the Suspicion-Agent, which integrates a ToM-aware planning approach that leverages higher-order ToM capabilities, considering not only what the opponent might do (first-order ToM) but also what the opponent believes the Suspicion-Agent will do (second-order ToM). Additionally, Yim et al. (2024) introduced a ToM planning method in the Guandan poker game, Liu et al. (2024) proposed an intention-guided mechanism, Xu et al. (2023) developed Probabilistic Graphical Modeling, and Zhang et al. (2024b) introduced K-Level-Reasoning, all utilizing ToM to enhance LLMs' strategic reasoning. Given the broad application of ToM, this paper leverages TMGBENCH to comprehensively evaluate LLMs' ability to employ first-order and second-order ToM reasoning techniques for strategic reasoning.

## 5  DISCUSSION

**Limitations.** Our TMGBENCH focuses on a very specific area within the vast domain of game theory, highlighting the fact that there is still a significant portion of game theory that lacks systematic exploration. While it may be infeasible or even impossible to develop a framework that is suitable for all types of games, we hope that benchmarks like TMGBENCH can help identify issues such as inherent imbalances and the non-robustness of LLMs' strategic reasoning abilities.

**Conclusion.** In this work, we introduce TMGBENCH, a benchmark for systematically evaluating the strategic reasoning abilities of LLMs by 2x2 matrix games. Based on Robinson-Goforth topology, we develop the classic setting tasks, and introduce various narratives based on story contexts generated by GPT-4o. By utilizing TMGBENCH, we can identify current flaws in LLMs' performance on these tasks, such as low accuracy rates and unstable inconsistency and bias degrees, even though the task difficulty is relatively moderate compared to many others. Additionally, when employing prompts to elicit their Theory-of-Mind thinkings on these tasks, some LLMs show improved performance, indicating that LLMs can, to some extent, master ToM and apply it in their reasoning processes. However, possessing first-order ToM abilities does not necessarily mean that LLMs will excel at mastering higher-order ToM. Furthermore, based on TMGBENCH, we introduce more forms of complex strategic reasoning tasks and pose a new challenge for LLMs.

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

## A    BRIEF INTRODUCTION TO STRATEGIC REASONING

**Definition.** Strategic reasoning (Gandhi et al., 2023; Zhang et al., 2024a) is a unique and sophisticated form of reasoning that focuses on making optimal decisions in multi-agent environments. It involves carefully selecting strategies by *anticipating the actions of others* and *understanding how one's choices will influence their responses*.

**Distinction.** What sets strategic reasoning apart is its *dynamic nature* and the *inherent uncertainty of adversarial actions*. Unlike other reasoning paradigms (commen sense reasoning, symbolic reasoning, casual reasoning, etc.), it demands a deep comprehension of ever-changing contexts and the ability to make rational, forward-thinking decisions based on the anticipated behaviors of others.

**Example.** In online advertising auctions (Edelman et al., 2007), advertisers compete for advertisement placements by bidding on specific audiences or keywords. Success depends on strategic reasoning, such as allocating budgets effectively, predicting competitors' bids, and targeting audiences where competition is lower. Advertisers must also optimize their advertisement quality to reduce costs while maintaining visibility. Since auctions are dynamic and often follow a second-price model (where the winner pays just above the second-highest bid), advertisers continuously adjust their strategies to balance cost and competitiveness. This interplay of decisions makes advertising auctions a prime example of strategic reasoning in real-world applications. Considering scenarios where strategic reasoning can be applied with LLMs, fields such as societal simulation, economic simulation, game theory, and gaming (Zhang et al., 2024a) are prominent areas that often require this capability.

**Significance.** Strategic reasoning is a cornerstone for enabling intelligent systems to operate effectively in complex, multi-agent environments. In the context of LLMs, equipping them with strategic reasoning capabilities extends their potential beyond static information retrieval or pattern recognition tasks. It allows LLMs to simulate realistic decision-making processes, navigate dynamic social or economic systems, and collaborate or compete with other agents. This is particularly crucial in applications such as policy design, automated negotiations, and multi-agent simulations, where understanding and anticipating others' behavior is essential for success. By fostering LLMs with strategic reasoning, we are able to bridge the gap between artificial intelligence and human-like adaptive decision-making, paving the way for more socially aware, context-sensitive, and intelligent systems that can tackle real-world challenges with greater precision and impact.

## B    BASIC THINGS ABOUT GAME THEORY

In this section, we discuss two fundamental concepts in game theory: dominant strategy and Nash equilibrium.

A dominant strategy is one that always provides a player with a payoff at least as high as any other strategy, regardless of the actions of other players. In other words, if a player has a dominant strategy, they will consistently choose it, as it either maximizes their payoff or does not reduce it, irrespective of the strategies chosen by others.

Nash equilibrium refers to a set of strategies, one for each player, where no player can benefit by unilaterally changing their strategy. At a Nash equilibrium, each player's strategy is the best response to the strategies of the other players. This means that if all players are following their Nash equilibrium strategies, no one has an incentive to deviate from their current strategy. It represents a stable state in the game where players' strategies are mutually optimal.

In many games, the dominant strategy equilibrium and Nash equilibrium may coincide, but not always. A dominant strategy equilibrium is a specific type of Nash equilibrium where each player has a strategy that is optimal regardless of others' strategies. However, in many cases, dominant strategies may not exist, requiring Nash equilibria to be identified through analysis and computation.

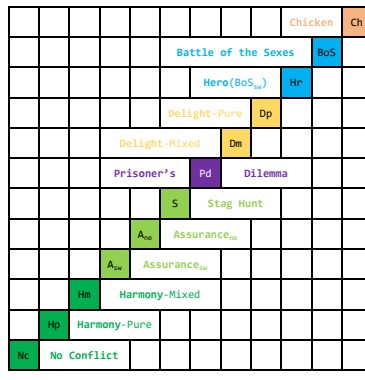

(a) Most Famous Games

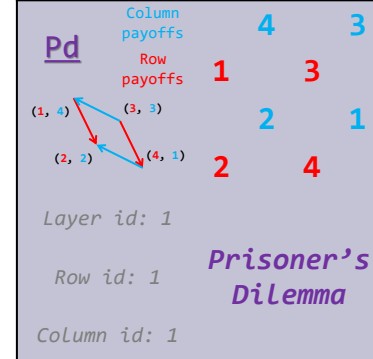

(b) Details in a Grid

Figure 9: The topology of the normal-form game system, which is presented by a square consisting of 12×12 grids. Figure 9a displays the position of the most famous games in the topology. In each grid, there are specific details of the game, which is shown in Figure 9b.

## C  2×2 MATRIX GAME

### C.1  DEFINITION

A normal-form game, commonly referred to as a 2×2 matrix game when involving two players each with two strategies, is a fundamental concept in game theory for representing strategic interactions. In this form, the game is depicted as a matrix, clearly outlining the players' strategies and corresponding payoffs. A typical 2×2 matrix game is structured as shown in Table 3.

Table 3: The form of typical 2×2 matrix games.

|  | Player B: Strategy 1 | Player B: Strategy 2 |
|---|---|---|
| Player A: Strategy 1 | (a, w) | (b, x) |
| Player A: Strategy 2 | (c, y) | (d, z) |

In this matrix, each cell represents the payoffs for both player A and player B, based on their chosen strategies. For instance, if player A selects strategy 1 and player B selects strategy 2, player A receives a payoff of $a$, while player B receives a payoff of $w$.

### C.2  TOPOLOGY

Game theory research often concentrates on the Prisoner's Dilemma and a few other symmetric games, even though most potential games are asymmetric, and many ordinal games involve ties.

The findings on the topology of ordinal normal-form games (Robinson & Goforth, 2005) provide an elegant framework for systematically studying these games, encompassing all equivalence classes in an ordinal sense (where "ordinal" refers to the ranking of payoffs rather than their specific values).

In this topological framework, as depicted in Figure 9, well-known games such as the Prisoner's Dilemma, Stag Hunt, Battle of the Sexes, and Chicken are all symmetric and situated on the counter-diagonal of a 12×12 grid. The remaining games are located in the other grids, each with a corresponding "sister game" that can be derived by reflecting across the counter-diagonal. A pair of sister games are identical when the roles of the two players are reversed.

Within each grid, basic information about the games in the equivalence classes is provided, including the family name and abbreviation, the payoff matrix, and the order graph, which illustrates the incentives for the row/column player to unilaterally change their choice for a higher payoff.

These 144 equivalence classes include 18 games with no equilibrium, 18 games with exactly two equilibria, and 108 games with a single equilibrium. Their distribution within the topology is symmetric across the counter-diagonal.

Figure 10: The distribution of games with 0, 1, or 2 Nash equilibria (a) is depicted according to the topology. Grids in grey indicate games with only 1 Nash equilibrium, while white grids represent games with no Nash equilibrium. Grids in other colours represent games with exactly 2 Nash equilibria. Text in blue/red indicates that the column/row player has a dominant strategy in the game, while white text signifies that both players have dominant strategies. In contrast, black text indicates that neither player has a dominant strategy.

### C.3   SOLUTION STRUCTURE

As previously mentioned, all games in the topological framework can be categorized into three distinct groups based on the number of Nash equilibria. If we consider Nash equilibrium as the solution to finding stable strategy combinations, Figure 10 illustrates the structure of these solutions.

In games with exactly one Nash equilibrium, at least one player (either the column player, row player, or both) has a dominant strategy, meaning they do not need to consider the other player's choice. These games are represented by grey or black grids.

Conversely, games with either 0 or 2 Nash equilibria share the characteristic that neither player has an unconditionally optimal choice, meaning no dominant strategies exist. However, in games with no Nash equilibrium (white grids), at least one player always has an incentive to unilaterally change their choice, regardless of the situation. In contrast, games with two Nash equilibria (orange, blue, or green grids) feature two stable strategy combinations.

Additionally, from a symmetry perspective, two sister games that are symmetric across the counter-diagonal belong to the same category and have identical Nash equilibria.

## D   MORE INFORMATION ABOUT OUR TMGBENCH

### D.1   GENERATION PIPELINE

In our study, we design an efficient dataset generation pipeline that leverages GPT-4o as the core to produce the entire dataset, with rigorous human quality reviews incorporated. The pipeline is organized into three carefully designed stages:

**Classic Game Construction.** Based on the topology of 2×2 games, we first introduce game descriptions for the payoff matrices of 144 game types, resulting in 144 classic games. An example of a classic game is shown below, which mirrors the structure of the Prisoner's Dilemma. These 144 classic games will serve as seed games, with their inherent game structures generalized into more diverse, story-based games.

---

**Example of classic game:** *classic/111*

[Scenario]
Player A and Player B are playing a game. Either of them has two choices, namely A1, A2/B1, B2. The payoff matrix of their different choice combinations is given below (larger number means higher payoff):

```
| A \ B | B1     | B2     |
|-------|-------|-------|
| A1    | 1 \ 4 | 3 \ 3 |
| A2    | 2 \ 2 | 4 \ 1 |
```

Both Player A and Player B are targeting maximizing their own payoff.
[/Scenario]

---

**Story-based Game Generation.** The aforementioned classic games offer a highly condensed mathematical representation of diverse game scenarios. However, in the real world, games often occur in complex social contexts involving various themes. To capture this complexity, we further designed *story-based games*, incorporating richer entities and more intricate game scenarios.

Specifically, we used synthetic data generation techniques and crafted detailed prompts to set the construction constraints for generating high-quality story-based games. Additionally, to enhance the realism of our game scenarios, we manually defined several thematic categories to guide the data synthesis process (see §D.3). Both the prompt constraints and thematic categories ensure the generated content aligns with the intended structure and thematic elements. An example of a generated story-based game is shown below, which follows the same game structure as the Prisoner's Dilemma and is presented within a new narrative context. As such, the story-based game `story-based/111_0` serves as a counterpart to the classic game `classic/111`. For each classic game, we generate five corresponding story-based games. The data synthesis prompt is as follows. The red text are the placeholders for the variables of the generation code, where "domain" indicates the topic we random-choose for the task, and "matrix_str" indicates the payoff matrix derived from the game structure we enumerate.

**Story-based Game Generation Prompt**

Please generate a game theory short story with the following requirements:
- Specific topic: {*domain*}
- There are two characters who may be in a situation of "cooperation" or "competition";
- Each character has 2 choices, and the combinations of their choices form 4 different scenarios;
- In these 4 scenarios, the two characters face different benefits/losses, which can be abstracted as different rewards they can obtain or different states they can achieve in each scenario;
- They each have a preference relationship for these rewards/states. We use numbers to represent the degree of preference, with 4 representing the most preferred and 1 the least preferred (i.e., preference degree 4>3>2>1);
- The payoff matrices for both characters can be abstracted and represented in one matrix, where A and B represent two characters and their choices are A1, A2/B1, B2. The respondent matrix is shown as below (the story you generate should have the same payoff structure as it):
{*matrix_str*}
Now please design a story that includes:
- Characters
- Each character's choices
- Characters' preferences for different scenarios
- Story description
Response format:
[Characters]
{
    "A": "...", "B": "..."
}
[/Characters]
[Choices]
{
    "A1": "...", "A2": "...", "B1": "...", "B2": "..."
}
[/Choices]
[Preferences]
Characters' preferences for different scenarios (4 most preferred, 1 least preferred):
{
    "A": {
        4: "...", 3: "...", 2: "...", 1: "..."
    },
    "B": {
        4: "...", 3: "...", 2: "...", 1: "..."
    }
}
[/Preferences]
[Payoff Matrix]
...
[/Payoff Matrix]
[Scenario]
...(to detailedly describe the situation, including the information of characters, choices, preferences and payoffs on different choice combinations)
[/Scenario]

Example of story-based task: *story-based/111_0.txt*

[Characters]
{
  "A": "Sarah, a small coffee shop owner",
  "B": "Mark, owner of a large coffee chain"
}
[/Characters]
[Choices]
{
  "A1": "Keep prices steady",
  "A2": "Lower prices",
  "B1": "Maintain current operations",
  "B2": "Expand aggressively in Sarah's area"
}
[/Choices]
[Preferences]
Characters' preferences for different scenarios (4 most preferred, 1 least preferred):
{
  "A": {
    4: "Lower prices while Mark expands (survive and gain market share)",
    3: "Keep prices steady while Mark maintains operations (stable coexistence)",
    2: "Lower prices while Mark maintains operations (unnecessary price war)",
    1: "Keep prices steady while Mark expands (lose customers)"
  },
  "B": {
    4: "Maintain operations while Sarah keeps prices steady (maximize profits)",
    3: "Expand while Sarah keeps prices steady (gain market share)",
    2: "Maintain operations while Sarah lowers prices (status quo)",
    1: "Expand while Sarah lowers prices (costly competition)"
  }
}
[/Preferences]
[Payoff Matrix]

```
| A \ B | B1      | B2      |
|-------|-------|-------|
| A1    | 1 \ 4 | 3 \ 3 |
| A2    | 2 \ 2 | 4 \ 1 |
```

[/Payoff Matrix]
[Scenario]
In a bustling city, Sarah runs a cozy, independent coffee shop that has been a local favorite for years.
Recently, Mark's large coffee chain has been eyeing expansion into Sarah's neighborhood, threatening her business. Sarah faces a critical decision: she can either keep her prices steady (A1) or lower them (A2) to attract more customers. Meanwhile, Mark must decide whether to maintain his current operations (B1) or aggressively expand into Sarah's area (B2). If Sarah keeps her prices steady and Mark maintains his current operations (A1, B1), Sarah struggles to compete (1) while Mark enjoys maximum profits (4). If Sarah lowers her prices and Mark stays put (A2, B1), both experience moderate success (2, 2) as Sarah attracts some new customers without directly competing with Mark. Should Mark decide to expand while Sarah keeps prices steady (A1, B2), both would face challenges but could coexist (3, 3) as Sarah retains loyal customers and Mark gains new ones. However, if Sarah lowers her prices as Mark expands (A2, B2), Sarah might survive and even gain market share (4), but Mark would face costly competition (1).
Their decisions will shape the local coffee market and determine the fate of Sarah's beloved shop.
[/Scenario]

**Quality Verification.** To ensure coherence and internal consistency in the generated games, we implement a multi-step generation strategy, incorporating meticulous human review. First, GPT-4o generates an initial draft of the story, which is then reviewed by a human for any inconsistencies or logical flaws. If the draft fails this review, GPT-4o is prompted to identify the problematic sections and apply a self-correction mechanism.

During the self-correction phase, GPT-4o analyzes the story for inconsistencies and revises the flawed sections. The revised version undergoes another round of human review. This iterative refinement process continues until the story meets the required quality standards.

If, after several rounds of regeneration, the story still contains significant issues or fails to meet the criteria, we may reject the output entirely. In such cases, the process is restarted from scratch with a new draft to ensure a fresh approach and to avoid perpetuating prior errors.

D.2    REASONING PROMPT USED

In this section, we present the prompts used by various reasoning methods. Notably, when invoking o1-mini to give response, we only use DA prompting, since the model are reported to perform reasoning internally and user should avoid' prompting like chain-of-thought.

---

*DA* prompting

**System:** You are a spectator, and you should answer question based on given senario.
**User:**
{*task description*} {*task question*}
Only give a block of python-style code containing your answer without any process. e.g.
'''python
answer = [("Ax", "By")] # list-type
'''

---

*CoT* prompting

**System:** You are a spectator, and you should answer question based on given senario.
**User:**
{*task description*} {*task question*}
Think step by step, and finally give a block of python-style code containing your answer. e.g.
'''python
answer = [("Ax", "By")] # list-type
'''

---

*FoToM* prompting

**System:** You are a spectator, and you should answer question based on given senario.
**User:**
{*task description*} {*task question*}
From A's perspective, try to figure out B's action and make choice. Then from B's perspective try to figure out A's action and make choice. Finally as a spectator, give a block of python-style code containing your answer. e.g.
'''python
answer = [("Ax", "By")] # list-type
'''

---

*SoToM* prompting

**System:** You are a spectator, and you should answer question based on given senario.
**User:**
*{task description} {task question}*
From A's perspective, try to figure out B's action, note that he may also reason based on your information or reasoning. Then from B's perspective try to figure out A's action, note that he may also reason based on your information or reasoning. Finally as a spectator, give a block of python-style code containing your answer. e.g.
'''python
answer = [("Ax", "By")] # list-type
'''

---

*reFoToM* prompting

**System:** You are a spectator, and you should answer question based on given senario.
**User:**
*{task description} {task question}*
From B's perspective, try to figure out A's action and make choice. Then from A's perspective try to figure out B's action and make choice. Finally as a spectator, give a block of python-style code containing your answer. e.g.
'''python
answer = [("Ax", "By")] # list-type
'''

---

*reSoToM* prompting

**System:** You are a spectator, and you should answer question based on given senario.
**User:**
*{task description} {task question}*
From B's perspective, try to figure out A's action, note that he may also reason based on your information or reasoning. Then from A's perspective try to figure out B's action, note that he may also reason based on your information or reasoning. Finally as a spectator, give a block of python-style code containing your answer. e.g.
'''python
answer = [("Ax", "By")] # list-type
'''

---

### D.3 BENCHMARK DIVERSITY

Our dataset is characterized by the diverse contexts encapsulated within the story-based tasks, a diversity that manifests across several dimensions.

Firstly, we have identified 20 distinct topics derived from everyday life scenarios where cooperation and competition are likely to occur. These topics align with situations commonly depicted in various game families. The distribution of story-based games across these 20 topics is visualized in Figure 11a.

The topics encompass a broad spectrum of fields, including Business, Ecology, Sports, Technology, Health Care, Politics, and more. Notably, Business constitutes the largest proportion of the dataset at 11.1%, while the remaining topics are more evenly distributed, with percentages generally ranging from approximately 1.4% to 7.9%.

Given the nature of these long-text reasoning tasks, the scenarios within our story-based games typically range from 200 to 450 words in length. As illustrated in Figure 11b, over 90% of scenario lengths fall within the 250 to 400-word interval. Additionally, we provide a scatter plot of scenario lengths by topic to further demonstrate the diversity of our generated dataset.

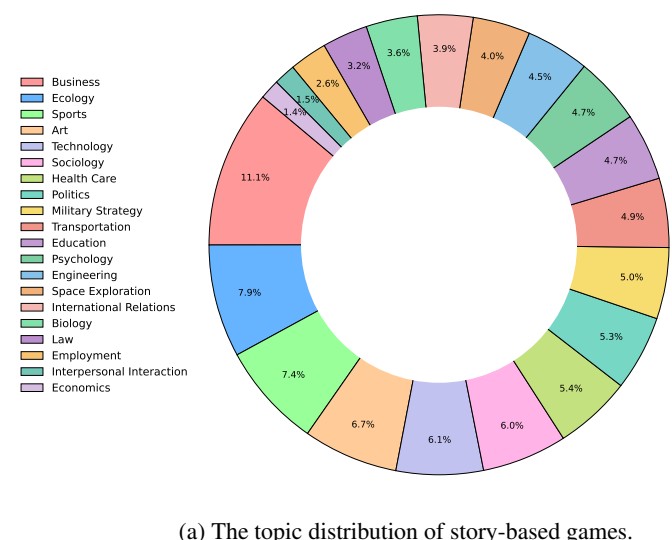

(a) The topic distribution of story-based games.

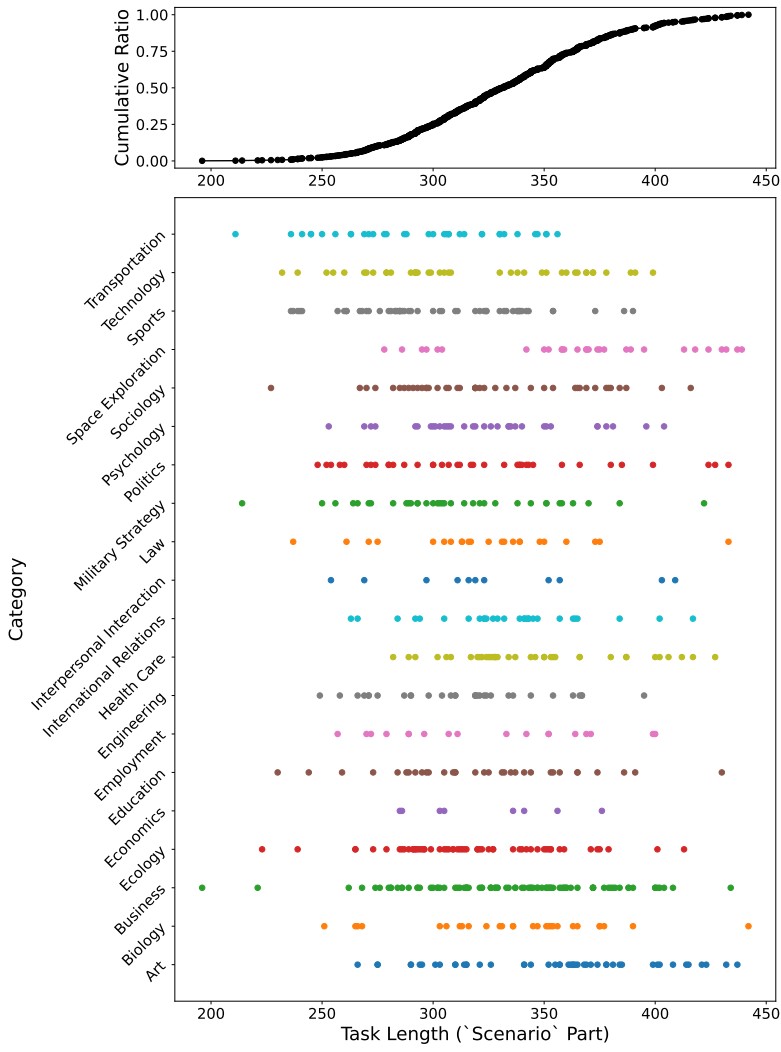

(b) Cumulative distribution of lengths by ratio and scatter plot of lengths by topic.

Figure 11: Statistical distribution of story-based games over 20 topics.

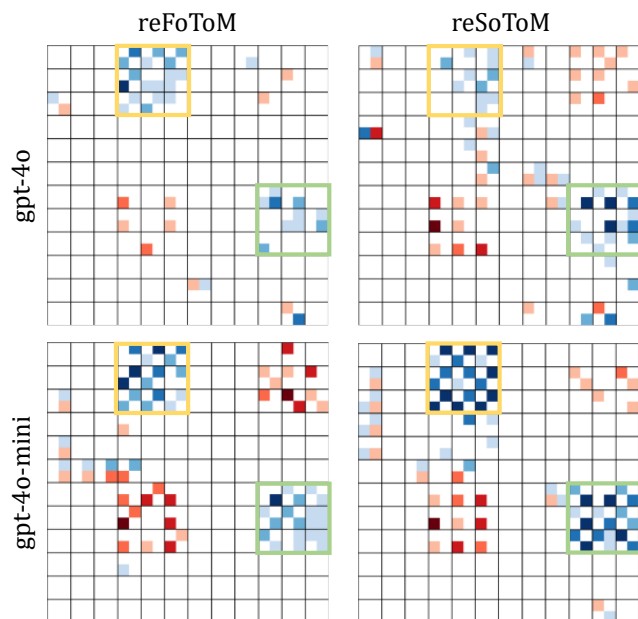

Figure 12: Inconsistency heat map of GPT series models using reFoToM and reSoToM prompting.

Table 4: The significance degree of top-tier GPT models performance. The larger value indicates the higher significance of the peculiar answering pattern. Near-zero value means no particular pattern. All values are expressed as percentages.

| Model | CoT | FoToM | ReFoToM | SoToM | ReSoToM |
|---|---|---|---|---|---|
| **gpt-4o** | 13.89 | 9.38 | 8.33 | 4.51 | 6.25 |
| **gpt-4o-mini** | 5.56 | 26.74 | 20.49 | 32.64 | 35.42 |

# E  ADDITIONAL RESULTS

## E.1  ASYMMETRIC INCONSISTENCY PATTERN

We show in Figure 12 that GPT series models still display similar pattern when using reFoToM and reSoToM prompting. Yellow-box areas and green-box areas display an asymmetric inconsistency pattern.

In order to further quantify how significant does the results display such pattern, we also propose a metric named significance degree which confined in [0, 1] and it is defined as

$$\text{SD} = \frac{1}{18} \sum_{i,j} \mathbb{I}\{\#\text{NE}(i,j) = 0\} \cdot \frac{1}{4}(\text{Freq}_{i,j,1} + \text{Freq}_{i,j,4} - \text{Freq}_{i,j,2} - \text{Freq}_{i,j,3}) * \text{S}(i,j)$$

where $\#\text{NE}(i,j)$ represents the number of Nash equilibria of the tasks of grid $(i,j)$, and $\text{S}(i,j)$ is decided by the area that the grid $(i,j)$ belongs to, having value 1 given the grid in the green area and value -1 given the grid in the yellow area.

We present the statistical results of LLMs' performance in Table 4, which show that the SD values for using ReFoToM are similar to those for FoToM, and the values for ReSoToM are close to those for SoToM.

Additionally, the results indicate that employing ToM helps gpt-4o reduce the asymmetric inconsistency pattern, while it conversely makes gpt-4o-mini more 'stubborn' in maintaining this pattern. Furthermore, higher-order ToM appears to have a stronger effect than first-order ToM.

## E.2 CASES OF FAILURE

In this section, we demonstrate two examples of the real response from gpt-4o-mini, which derived from a pair of symmetric data points (the grid (row 3, column 4) and the grid (row 9, column 10)) that theoretically have the same answer and similar reasoning pattern. We use the red coloured texts to indicate the mistaken parts of the reasoning processes.

---

**row 3 column 4**

To determine the most likely choice combinations for Players A and B, we can analyze the payoff matrix step by step.
1. **Identify the Payoffs**: The payoff matrix is as follows:

```
| A \ B | B1      | B2      |
|-------|-------|-------|
| A1    | 3 \ 1 | 1 \ 4 |
| A2    | 2 \ 3 | 4 \ 2 |
```

2. **Player A's Choices**:
   - If Player B chooses B1:
     - A1 gives Player A a payoff of 3.
     - A2 gives Player A a payoff of 2.
   - If Player B chooses B2:
     - A1 gives Player A a payoff of 1.
     - A2 gives Player A a payoff of 4.
   From Player A's perspective:
     - Against B1, A1 (3) is better than A2 (2).
     - Against B2, A2 (4) is better than A1 (1).
3. **Player B's Choices**:
   - If Player A chooses A1:
     - B1 gives Player B a payoff of 1.
     - B2 gives Player B a payoff of 4.
   - If Player A chooses A2:
     - B1 gives Player B a payoff of 3.
     - B2 gives Player B a payoff of 2.
   From Player B's perspective:
     - Against A1, B2 (4) is better than B1 (1).
     - Against A2, B1 (3) is better than B2 (2).
4. **Best Responses**:
   - If Player A chooses A1, Player B will choose B2 (since 4 >1).
   - If Player A chooses A2, Player B will choose B1 (since 3 >2).
   - If Player B chooses B1, Player A will choose A1 (since 3 >2).
   - If Player B chooses B2, Player A will choose A2 (since 4 >1).
5. **Nash Equilibria**:
   - The combination (A1, B2) gives payoffs (3, 4).
   - The combination (A2, B1) gives payoffs (2, 3).
6. **Conclusion**:
   - The most likely choice combinations are (A1, B2) and (A2, B1). Since both combinations are viable and lead to different outcomes, we conclude that both are possible.
Thus, the final answer in the required format is:
```python
answer = [("A1", "B2"), ("A2", "B1")] # list-type
```

---

**row 9 column 10**

To determine the most likely choice combinations for Players A and B, we can analyze the payoff matrix step by step.
1. **Identify the Payoffs**:
   - For Player A:
     - If A chooses A1:
       - Against B1: Payoff = 2
       - Against B2: Payoff = 4
     - If A chooses A2:
       - Against B1: Payoff = 3
       - Against B2: Payoff = 1
   - For Player B:
     - If B chooses B1:
       - Against A1: Payoff = 4
       - Against A2: Payoff = 2
     - If B chooses B2:
       - Against A1: Payoff = 1
       - Against A2: Payoff = 3
2. **Best Responses**:
   - If Player B chooses B1:
     - Player A's best response is A2 (3 >2).
   - If Player B chooses B2:
     - Player A's best response is A1 (4 >1).
   - If Player A chooses A1:
     - Player B's best response is B1 (4 >1).
   - If Player A chooses A2:
     - Player B's best response is B2 (3 >2).
3. **Nash Equilibria**:
   - A Nash Equilibrium occurs when both players are choosing their best responses to each other.
   - From the analysis:
     - (A1, B1) is a Nash Equilibrium because A's best response to B1 is A2, but B's best response to A1 is B1.
     - (A2, B2) is also a Nash Equilibrium because A's best response to B2 is A1, but B's best response to A2 is B2.
4. **Conclusion**:
   - The most likely choice combinations are (A1, B1) and (A2, B2) since both are Nash Equilibria.
Now, we can represent this in the required Python-style code:
```python
answer = [("A1", "B1"), ("A2", "B2")] # list-type
```

# F About Complex Form Game

## F.1 Example of Different Complex Form

In this section, we use visualized example (referred to Figure 13) to illustrate different kinds of complex forms.

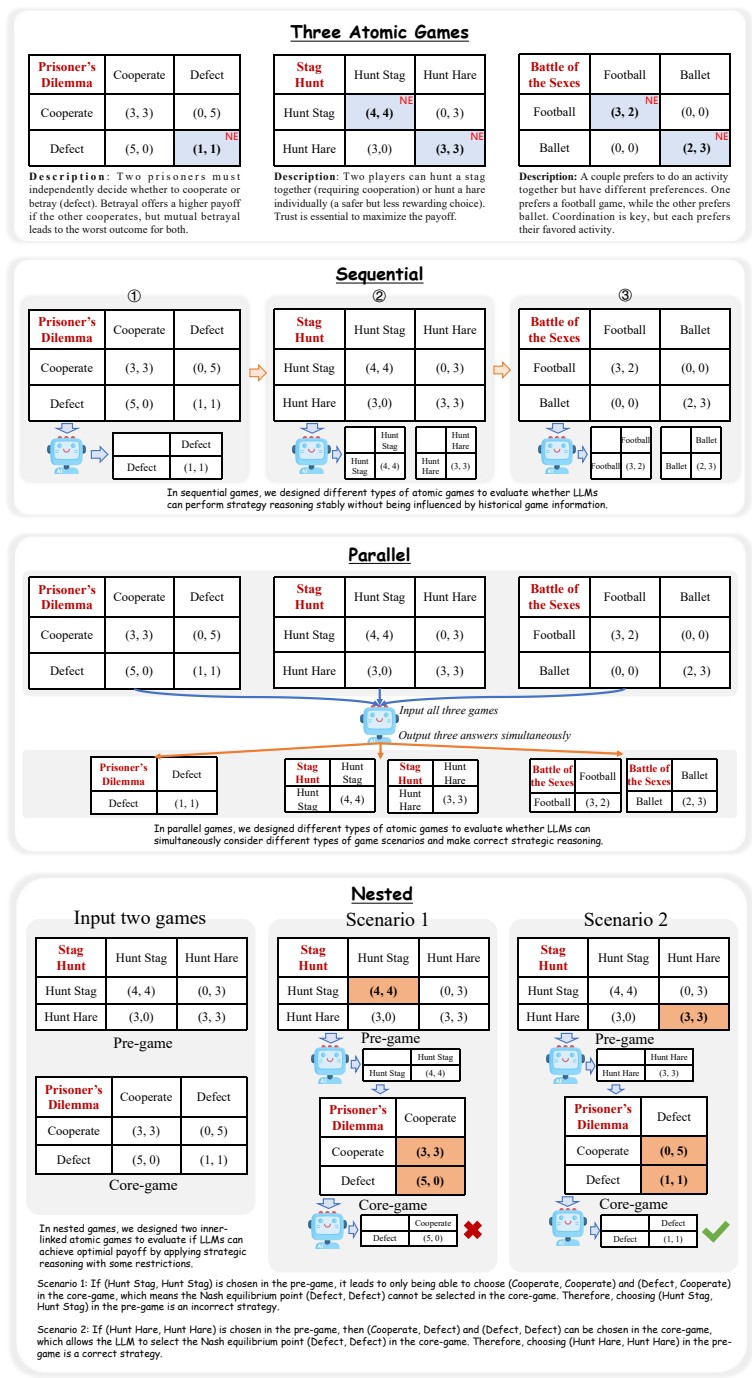

Figure 13: An example of how we build sequential, parallel, and nested game from some of the atomic games in TMGBench.

## F.2 NESTED FORM GAME

In this section, we demonstrate the template we use for generating nested form games. The text in red are the placeholders for the variables of the generation codes.

---

**template of a kind of nested form game**

<Pre-Game >
{pre_game}
<Core-Game >
{core_game}
[Question]
Player A and B are facing the two games, the pre-game and the core-game.
Note that their final goal is to maximize own payoff first in the core Game, then in the pre-game.
Additionally, {restricted_player} is attached with an restriction that if the situation of the pre-game is {restricted_situation}, then he can not choose action {restricted_choice}.
What is/are the most possible choice combination(s) of the pre-game ultimately? (when all choice combinations have equal possibility, the answer should contain nothing)
[/Question]

---

After a nested form game is generated through our template, we still need to check if the Nash equilibria of the pre-game changes after the restriction from the core game. If the set of Nash equilibria does change, then we use this as a piece of data to evaluate LLMs, observing if they can observe such a violation of original NEs' structure.

## F.3 SIGNIFICANCE OF ATOMIC GAMES AND COMPLEX-FORM GAMES

Our evaluation of complex-form games serves as a test of whether LLMs can solve real-world problems with higher complexity, rather than merely solving a single atomic game in isolation. The atomic games in TMGBENCH represent the *primary components* of complex real-world social scenarios (Gintis, 2014). In addition to the typical combinations found in temporal contexts (i.e., sequential games) or spatial contexts (i.e., parallel games), which require reasoning and independent decision-making, Tsebelis (1990) introduced a concept known as *nested games*, where two or more games are inner-linked. This type of game composition often arises in real-world domains such as politics and economics, where decisions made in one arena can influence or constrain decisions in another.

