# OpenReview forum: "TMGBench: A Systematic Game Benchmark for Evaluating Strategic Reasoning Abilities of LLMs"
_ICLR.cc/2025/Conference — Submitted to ICLR 2025_

### Official Review · Reviewer_XhyT · 2024-10-20

**Soundness:** 3
**Presentation:** 2
**Contribution:** 2
**Rating:** 5
**Confidence:** 4

**Summary:**

The paper presents a benchmark for strategic reasoning comprised of all 2x2 game ordinal payoff arrangements. Additional evaluation capabilities include testing agents when reasoning on compositions of these games (in parallel, sequentially, or where one game influence the choices in a subsequent game) and reframing the games in story-based scenarios. Evaluations study open and closed source LLMs on this benchmark, assessing: how well they produce optimal choices, the extent to which they exhibit asymmetrically biased responses when payoff matrices are flipped, and using theory of mind to improve performance. The results demonstrate that existing LLMs do not saturate the benchmark, have varying degrees of bias based on the payoff structure and story framing, and struggle to leverage theory of mind to improve results.

**Strengths:**

# originality
Modest.

Evaluating LLMs in strategic reasoning games is a thoroughly investigated topic (as attested by the related work). Examining anti-symmetric reasoning patterns is a question I have not seen probed before and important to consider for this setting in general.

# quality
Modest.

Experiments demonstrate the benchmark can find differences among LLMs. Models fail to saturate the success criteria, particularly for more stringent requirements like perfect answering or demonstrating theory of mind. Biases based on the generated stories show there is clear room for improving LLM context sensitivity, however it is not clear how much this could be mitigated by different prompts for the strategic reasoning (a dimension not explored in the paper).

# clarity
Modest.

The introduction was vague and hard to follow without reading the rest of the paper. Experiments are documented well. Some figures were hard to parse or could use a different presentation (notes below).

# significance
Modest.

There are numerous evaluations for strategic reasoning in game theoretic games. This focuses on 2x2 games, omitting multi-agent agents or repeated/multi-turn games (excepting the composite games tested). The paper will be of some interest to the community focusing on this subset of LLM capabilities.

**Weaknesses:**

Note: These weaknesses are phrased as questions to facilitate discussion.

# originality
How do the games in this benchmark cover those not covered in the "game theory" subsection of the cited paper "A Survey of Strategic Reasoning with Large Language Models"? Or for the "Societal Behavior" examples that include theory of mind?


# quality

The experiments should include statistical tests when claiming differences among model types. At least in the cases where multiple runs were possible or multiple scenarios are being aggregated (for example, in Table 1 and Figure 5). Many claims seem plausible, but the tests are there to provide rigor.

The paper would benefit from evaluating the concern stated in the introduction that there is scenario leakage of common game forms. Was there evidence of scenario leakage based on the games in Robinson-Goforth topology results? Do the games most likely to be leaked (like Prisoner's Dilemma) demonstrate substantial performance differences relative to other games?


# clarity

The introduction could be clearer on details later explained in the paper. Examples:
- "performance variability marked by coefficients" - Coefficients of what?
- "marked by an asymmetric pattern" - What asymmetric pattern?

Figure 6 is hard to read. It might be better plotted by showing the differences in scores between the classic and story-based settings instead.

# significance

What are the key insights we learn from TGMBench that were not revealed in prior benchmarks? This is not very clearly articulated in the paper and would help establish it's originality and significance. As TGMBench is a benchmark, the value it provides derives in exposing LLM capabilities that are not already apparent in alternatives.

**Questions:**

See above.

---

> ### Author Response · Authors · 2024-11-17
> **Thanks for your careful review! (1/2)**
>
> We thank you for genuinely providing valuable comments on our paper. We will address your concerns one by one.
>
> **Originality:**
>
> We appreciate your question regarding the games covered in our benchmark relative to *“A Survey of Strategic Reasoning with Large Language Models”* and examples involving Theory of Mind (ToM). We acknowledge that strategic reasoning is a vast domain with many forms, and we chose to focus on a specific subset of games (within the subset, the atomic games have a common form but different configurations and descriptions), particularly those resembling scenarios like the Prisoner’s Dilemma, Stag Hunt, etc., where strategic reasoning is fundamental. These games are a subset of the larger landscape of strategic reasoning tasks, and even within this subset, we found that current LLMs struggle with reasoning consistently and accurately.
>
> In summary, compared to previous work, our benchmark: (1) considers a more comprehensive set of 144 types of 2x2 games; (2) explores different contextual framings of the same game structure in greater depth; and (3) introduces three novel complex game forms—sequential, parallel, and nested—based on atomic games, which were not designed in prior studies.
>
> By focusing on this category of problems, we aim to reveal specific deficiencies in LLMs’ strategic reasoning abilities, such as the “asymmetric pattern” observed in symmetric games, which has not been well-studied in prior benchmarks. While other benchmarks may explore broader game-theoretic concepts, our contribution lies in systematically evaluating LLMs on a targeted and relevant class of strategic reasoning tasks.
>
> ---
>
> **Quality:**
>
> Thank you for suggesting the incorporation of statistical tests to assess differences between models. We fully agree that this would enhance the rigor of the paper and strengthen our findings. In future revisions and subsequent work, we plan to include such statistical analyses.
>
> We also share your observation that some games, such as the Prisoner’s Dilemma, might be more commonly exposed to LLMs during training, potentially leading to better performance on those games. To investigate this, we collected experimental data and computed metrics for two distinct sets of games: the counter-diagonal set (which includes well-known games like the Prisoner’s Dilemma, Stag Hunt, and Battle of the Sexes) and the non-counter-diagonal set. The results are summarized in the table below (using DA prompting):
>
> | Model             | PAR (↑, Famous/Total) | ID (↓, Famous/Total) | BD (↓, Famous/Total) |
> |--------------------|-----------------------|-----------------------|-----------------------|
> | gpt-4o            | 46.88/52.08           | 16.93/16.81           | 25.52/28.49           |
> | gpt-4o-mini       | 29.17/14.06           | 27.99/39.52           | 61.20/56.21           |
> | gpt-3.5-turbo     | 40.63/30.21           | 25.39/27.64           | 50.78/50.15           |
> | claude-3-5-sonnet | 45.83/59.38           | 16.67/14.79           | 33.33/27.76           |
> | claude-3-haiku    | 29.17/24.31           | 33.33/39.58           | 83.33/72.22           |
> | Llama-3.1-70B     | 29.17/13.02           | 12.50/36.15           | 25.00/40.71           |
> | Llama-3.1-8B      | 12.50/18.75           | 43.75/38.49           | 87.50/81.32           |
> | Qwen2-72B         | 33.33/43.06           | 25.00/26.30           | 33.33/35.59           |
>
> From the table, we observe that for advanced models such as **gpt-4o**, **claude-3-5-sonnet**, and **Qwen2-72B**, performance on the famous set of games does not consistently surpass (and in some cases is lower than) performance across all games. Conversely, for models like **Llama-3.1-70B** and **gpt-4o-mini**, the famous game set appears to be relatively easier. This is a fascinating finding and may indicate potential training data leakage for the more well-known games.
>
> We acknowledge that this raises a significant and valuable research question, and we plan to explore this direction further in future work. Your observation has been instrumental in highlighting an area that warrants deeper investigation. Thank you for bringing this to our attention.
>
> ---
>
> **Clarity:**
>
> We appreciate your suggestions for improving clarity. We revise the introduction in our new revision on line 91 and line 94 respectively. Also we include another appendix section to provide clearer explanations for some terms.
> Regarding Figure 6, we will explore alternative visualizations that better highlight the differences between the classic and story-based settings, as suggested.

---

> > ### Author Response · Authors · 2024-11-17
> > **Thanks for your careful review! (2/2)**
> >
> > **Significance:**
> >
> > Our contributions are summarized as follows, and we look forward to receiving your recognition.
> >
> > - **Granular Evaluation:** Our benchmark provides a finer-grained evaluation of current LLMs’ performance on strategic reasoning tasks. We specifically highlight the “asymmetric pattern” observed in pairs of symmetric games, a phenomenon that reveals gaps in LLMs’ reasoning abilities and demonstrates the difficulty LLMs face in applying general strategies across different scenarios.
> >
> > - **Complex Game Compositions:** We also introduce the concept of more complex real-world games that can be constructed by combining atomic games in various forms. We demonstrate how these combinations create more challenging scenarios for LLMs and evaluate the performance of state-of-the-art LLMs in these more complex environments. Our work proposes several ways to combine simpler games into more complex scenarios, which opens up avenues for testing LLMs on a wider range of strategic reasoning tasks.
> >
> > Moreover, we would like to reveal TMGBench’s potential value for future LLM design. Our benchmark identifies key areas where LLMs need improvement:
> >
> > - **Long-Context Reasoning:** Our experiments identify mistakes in the internal process of LLMs’ strategic reasoning, which calls for better long-context reasoning ability.
> > - **Theory of Mind:** Our results highlight the need for more robust theory of mind capabilities, as LLMs still exhibit drawbacks when applying ToM (e.g., inconsistency, asymmetric patterns).
> > - **Understanding Multi-Participant Social Scenarios:** Through our data generation process, we observe that LLMs sometimes struggle to accurately understand social scenarios involving both conflict and cooperation.
> >
> > Overall, we see TMGBench as both a diagnostic tool for evaluating LLMs’ strategic reasoning capabilities and a guide for enhancing future LLMs with more complex and robust reasoning abilities.
> >
> > We look forward to engaging in further discussions with you and receiving your additional guidance and feedback. Thank you!

---

> > > ### Comment · Reviewer_XhyT · 2024-11-19
> > >
> > > Thank you for providing detailed responses to my review and offering the new analysis on the issue of data leakage. The details help clarify some points that were not well expressed in the first version of the paper.
> > >
> > > These replies have addressed aspects that were unclear in the presentation but have not substantially shifted my confidence in the results being presented due to the absence of statistical tests and apparent data leakage concerns revealing that the benchmark may be fundamentally subject to contamination.
> > >
> > > >In future revisions and subsequent work, we plan to include such statistical analyses.
> > >
> > > I look forward to seeing the results!
> > >
> > > >Regarding Figure 6, we will explore alternative visualizations that better highlight the differences between the classic and story-based settings, as suggested.
> > >
> > > I look forward to seeing this as well.
> > >
> > > > From the table, we observe that for advanced models such as gpt-4o, claude-3-5-sonnet, and Qwen2-72B, performance on the famous set of games does not consistently surpass (and in some cases is lower than) performance across all games. Conversely, for models like Llama-3.1-70B and gpt-4o-mini, the famous game set appears to be relatively easier. This is a fascinating finding and may indicate potential training data leakage for the more well-known games.
> > >
> > > This is certainly interesting. To me this makes it clear that the benchmark needs some other means of organization, as the groups being aggregated may be mixing categories (like the famous vs not distinction here).
> > >
> > > Is there any way to show that the particular groups of games used are "good" groupings? This may be too vague to really answer.

---

> > > > ### Author Response · Authors · 2024-11-27
> > > > **Thanks for your follow-up comment!**
> > > >
> > > > Thank you for the follow-up comment!  We address the following concerns:
> > > >
> > > > **Concern 1: Absence of Statistical Analysis**
> > > >
> > > > We sincerely appreciate the reviewer’s concern regarding the absence of statistical tests in our work. We acknowledge the importance of incorporating rigorous statistical analyses to enhance the robustness of our findings. While a comprehensive statistical evaluation of all comparisons is beyond the scope of this paper, we are willing to include detailed statistical testing in future work. To address your suggestion, we have conducted an example statistical analysis here, focusing on model comparisons.
> > > >
> > > > As noted in line 348, *gpt-4o, gpt-4o-mini, and claude-3-5-sonnet are more capable compared to other models*. To further substantiate this claim, we performed a **Friedman rank test** to compare the performance of stronger models against weaker models:
> > > >
> > > > **Hypothesis H_0:** Model A and Model B have no significant performance difference over 144 games with CoT prompting.
> > > > **Hypothesis H_1:** Model A and Model B have a significant performance difference over 144 games with CoT prompting.
> > > >
> > > > The results of the test are presented in the table below:
> > > >
> > > > | Model A     | Model B           | X²    | Accept H₀ (F ≈ 3.06) |
> > > > | :---------- | :---------------- | :---- | :------------------- |
> > > > | gpt-4o      | claude-3-5-sonnet | 0.44  | Yes                  |
> > > > | gpt-4o      | gpt-4o-mini       | 11.11 | No                   |
> > > > | gpt-4o      | gpt-3.5-turbo     | 75.11 | No                   |
> > > > | gpt-4o      | Qwen2-72B         | 29.34 | No                   |
> > > > | gpt-4o-mini | gpt-3.5-turbo     | 43.34 | No                   |
> > > >
> > > > From the results, we observe that the null hypothesis (H₀) is accepted for the pair (gpt-4o, claude-3-5-sonnet), indicating no significant performance difference between these two models. However, for other model pairs, such as (gpt-4o-mini, gpt-3.5-turbo), the null hypothesis is rejected, suggesting a significant performance difference.
> > > >
> > > > Although we cannot perform an exhaustive statistical analysis for all comparisons in this paper, we greatly value your feedback on the need for rigorous testing. We will incorporate more comprehensive statistical evaluations in our future work to further enhance the robustness and credibility of our findings.
> > > >
> > > > **Concern 2: Potential Data Contamination**
> > > >
> > > > Thank you for your insightful comments on potential data contamination. In order to deeper resolve your concern on data contamination, we do additional analysis on our dataset:
> > > >
> > > > 1. **Source Analysis:**
> > > > 	Our dataset is **synthetic** and **template-based**, which significantly reduces the likelihood of explicit contamination. Since the data is generated using predefined templates and rules, there is a very low probability that any real-world data or previously encountered examples could seep into the dataset. This template-based approach helps maintain consistency across the examples, ensuring that the LLMs are evaluated purely on their strategic reasoning capabilities rather than being influenced by previously seen examples.
> > > >
> > > > 2. **Perplexity (PPL) Analysis:**
> > > > 	We performed a **perplexity analysis** on the dataset, testing it with those open-source models. Our findings show that the PPL values are within reasonable ranges, indicating that the dataset does not exhibit typical signs of contamination:
> > > >
> > > > 	- For **shorter classic data points**, the PPL values range from **8 to 10**:
> > > >
> > > > 	| Model         | Avg    | Std    |
> > > > 	| ------------- | ------ | ------ |
> > > > 	| Llama-3.1-8B  | 8.9532 | 0.1733 |
> > > > 	| Llama-3.1-70B | 8.0626 | 0.1552 |
> > > > 	| Qwen2-72B     | 9.0817 | 0.0913 |
> > > >
> > > > 	- For **longer story-based data points**, the PPL values range from **3 to 6**:
> > > >
> > > > 	| Model         | Avg    | Std    |
> > > > 	| ------------- | ------ | ------ |
> > > > 	| Llama-3.1-8B  | 5.0239 | 0.2308 |
> > > > 	| Llama-3.1-70B | 4.2513 | 0.1921 |
> > > > 	| Qwen2-72B     | 3.8923 | 0.1389 |
> > > >
> > > > 	These PPL values suggest that the data is not overly predictable, indicating that the dataset is not contaminated. The variation in perplexity across different models demonstrates that there is no significant bias or discrepancy in predicting any of the data points. This further suggests that there are no obvious signs of data leakage. However, while we have not observed any direct evidence of leakage, we acknowledge that we cannot fully rule out the possibility of subtle contamination.
> > > >
> > > > However, we might be not able to address the concern of *alternative visualization of Figure 6* right now, sorry for that. (If you have some ideas on that, we will be very happy to discuss about it.)
> > > >
> > > > Again, we still look forward to engaging in more discussions with you and receiving your additional guidance and feedback. Thank you!

---

> > > > > ### Comment · Reviewer_XhyT · 2024-11-29
> > > > >
> > > > > Thank you for the additional efforts! It looks like the main concerns I have require more time to be developed, so I will maintain my current score.
> > > > >
> > > > >
> > > > >
> > > > > >Conversely, for models like Llama-3.1-70B and gpt-4o-mini, the famous game set appears to be relatively easier.
> > > > >
> > > > > I think there may be some confusion around the data contamination. I'm not sure how the PPL scores above would address the problem of data leakage on those games. I may be misunderstanding the details of your analysis above and how that connects to the observations about the famous games being easier. Does the new PPL data indicate that the famous games have equal perplexity scores as the non-famous? Or would the famous games perhaps be systemically easier in general?

---

> > > > > > ### Author Response · Authors · 2024-11-29
> > > > > > **Thanks for your further comment!**
> > > > > >
> > > > > > Thank you so much for your further comment! We greatly appreciate your engagement and the opportunity to clarify our analysis.
> > > > > >
> > > > > > To address your concerns, we would like to emphasize that the conclusions drawn from our source and perplexity (PPL) analyses are not contradictory to the observations about famous vs. unfamous games——our TMGBench has a relatively low risk of data leakage, as outlined below.
> > > > > >
> > > > > > First, the dataset of TMGBench features on the synthetic context and we use a standardized template to develop it, and even for the **story-based** part, they have such long context (**200\~450 tokens** per data point, as shown in **Figure 11** in the paper). This substantially lowers the likelihood of any entire paragraph being exposed to LLMs during training.
> > > > > >
> > > > > > Also, for the **classic** part, from our original experiments and additional ablations, it is possible that some games have been exposed to LLMs, but **it will not be a risk**, and we provide some reasoning as below:
> > > > > >
> > > > > > 1. The normal level of perplexity tells us that there is **low possibility** of exposing an **entire same** data point of our TMGBench to LLMs, for the definition of perplexity is to measure to what extent LLM can correctly predict next token based on the preceding context.
> > > > > > 2. Some finding of the paper, along with the additional finding on famous vs. unfamous games, indicate that some LLMs might be **familiar** with some of the games, while this kind of **familiarity**, is not derived from the leakage of our data point. Instead, it is most probably originated from their **emerging ability** that they utilize to apply such **familiar** knowledge to do better in reasoning. For example, the training corpus of a LLM may include content which indirectly related to a famous game like *The Prisoner’s Dilemma* rather than an unfamous games, so LLMs have a chance to know more information about the game with similar game structure. (This is similar to solve a math problem or a card game, both need some pre-knowledge/experience which can boost ones‘ performance, right?)
> > > > > > 3. Actually, in many prior studies [1, 2, 3, 4, 5] , some classic games (not necessarily bi-matrix games, but well-known in game theory or economics) are still being employed to evaluate if LLMs can conduct strategic reasoning like humans or even perform better. Compared to these work, TMGBench have a lower risk of data leakage because our data points are **much longer** and we use a **synthetic** method which incorporates **much more variables**.
> > > > > >
> > > > > > In conclusion, while famous games might be easier for LLMs due to their inherent familiarity, this familiarity is not a consequence of TMGBench data leakage. Instead, it reflects the models' ability to leverage prior knowledge, further demonstrating the value of TMGBench in assessing strategic reasoning.
> > > > > >
> > > > > > Once again, thank you for your feedback, and we value your continued engagement with our work!
> > > > > >
> > > > > > ---
> > > > > >
> > > > > > [1] Aher, Gati V., Rosa I. Arriaga, and Adam Tauman Kalai. "Using large language models to simulate multiple humans and replicate human subject studies." *International Conference on Machine Learning*. PMLR, 2023.
> > > > > >
> > > > > > [2] Horton, John J. *Large language models as simulated economic agents: What can we learn from homo silicus?*. No. w31122. National Bureau of Economic Research, 2023.
> > > > > >
> > > > > > [3] Guo, Jiaxian, et al. "Suspicion-agent: Playing imperfect information games with theory of mind aware gpt-4." *arXiv preprint arXiv:2309.17277* (2023).
> > > > > >
> > > > > > [4] Duan, Jinhao, et al. "Gtbench: Uncovering the strategic reasoning limitations of llms via game-theoretic evaluations." *arXiv preprint arXiv:2402.12348* (2024).
> > > > > >
> > > > > > [5] Mei, Qiaozhu, et al. "A Turing test of whether AI chatbots are behaviorally similar to humans." *Proceedings of the National Academy of Sciences* 121.9 (2024): e2313925121.

---

> > > > > > > ### Author Response · Authors · 2024-12-02
> > > > > > > **Thank you for your review and feedback**
> > > > > > >
> > > > > > > Dear reviewer XhyT,
> > > > > > >
> > > > > > > Thank you for your review and constructive feedback! We hope that our extra responses can resolve the rest of the concerns you had. We have already thoroughly explained our viewpoint about the issue of data contamination in the comments and included sufficient evidential references. Please feel free to share any additional comments or suggestions. We greatly appreciate your thorough review and continued support.
> > > > > > >
> > > > > > > Best,
> > > > > > >
> > > > > > > Paper 6932 Authors

---

> > > > > > > > ### Author Response · Authors · 2024-12-03
> > > > > > > > **Look forward to your new feedback**
> > > > > > > >
> > > > > > > > Dear reviewer XhyT,
> > > > > > > >
> > > > > > > > We are very concerned whether our response has addressed your concerns and look forward to your new feedback.
> > > > > > > >
> > > > > > > > Best,
> > > > > > > >
> > > > > > > > Paper 6932 Authors

---

> > > > > > > > > ### Author Response · Authors · 2024-12-03
> > > > > > > > > **seek for latest feedback**
> > > > > > > > >
> > > > > > > > > Dear reviewer XhyT,
> > > > > > > > >
> > > > > > > > > Given the rebuttal deadline, we kindly request your latest feedback at your earliest convenience. Thank you for your understanding and prompt attention to this matter.
> > > > > > > > >
> > > > > > > > > Best,
> > > > > > > > >
> > > > > > > > > Paper 6932 Authors

---

### Official Review · Reviewer_SyXy · 2024-11-04

**Soundness:** 4
**Presentation:** 4
**Contribution:** 4
**Rating:** 8
**Confidence:** 2

**Summary:**

This paper proposes a benchmark TMGBENCH. TMGBENCH incorporates 144 game types based on the Robinson-Goforth topology of 2×2 games and provides three forms (sequential, parallel, and nested) to construct more complex games using those 144 game types. Several LLMs were compared on the benchmark using several quantified metrics to identify their strengths and weaknesses.

**Strengths:**

- The paper is very well-written.
- Objectives are clear, and how those objectives are achieved by this work is well demonstrated.
- Quantified metrics and visualisations have been used to compare LLMs on different tasks to assess their capabilities.
- Extensive experiments were conducted to exam the failure cases and the effect of ToM.
- Limitations were also discussed.
- Generation pipeline was demonstrated in Appendix.
Overall, the reviewer quite enjoyed reading this paper.

**Weaknesses:**

No particular weakness was identified by the reviewer. The reviewer is not an expert in game theory or reasoning. It is quite likely that the reviewer is unfamiliar with some pieces of related work or crucial part of this work.

**Questions:**

It is stated that “Theoretically, using these atomic games, we can expand the framework to generate infinitely many increasingly complex game forms.” However, standard answers are required to compute the inconsistency map. The reviewer wonders how to obtain the standard answers to newly generated games?

---

> ### Author Response · Authors · 2024-11-17
> **Thanks for your careful review!**
>
> We thank you for giving a high degree of recognition to our work and for effectively summarizing our core contributions.
>
> One regret of this work is that we were unable to provide a more detailed background in the main text, due to the page limit, to help readers unfamiliar with game theory quickly grasp the designs of our TMGBench. We will revise the paper to include more introductory parts in the appendix to clarify some concepts.
>
> We address your concern as follows:
>
> **Question**: How to compute the standard answers to complex form games?
>
> **Response**: We present three kinds of complex forms in our work: sequential, parallel, and nested. In the sequential and parallel forms, the atomic games are independent of each other, so we directly compute the standard answer for each atomic game using the conclusions from the Robinson-Goforth topology. However, in the nested form (we explore the 2-folded nested form in our work), we compute the conditional Nash equilibrium using the functions below (This means that all answers can be automatically computed based on the rules, ensuring their strict correctness):
> ```
> def get_Nash_equilibrium(pA, pB, ban=None, banp=None):
>     # pA: player A's payoff matrix
>     # pB: player B's payoff matrix
>     # ban: the restricted situation
>     # banp: the restricted player
>     Nash_equilibrium_choice, Nash_equilibrium_result = [], []
>     for row in range(2):
>         for column in range(2):
>             alter_row, alter_column = 1 - row, 1 - column
>             if ban is not None and (row + 1, column + 1) == ban: continue
>             if (banp == "A" and (alter_row + 1, column + 1) == ban or pA[row][column] >= pA[alter_row][column]) \
>             and (banp == "B" and (row + 1, alter_column + 1) == ban or pB[row][column] >= pB[row][alter_column]):
>                 Nash_equilibrium_choice.append((f"A{row + 1}", f"B{column + 1}"))
>                 Nash_equilibrium_result.append((pA[row][column], pB[row][column]))
>     return Nash_equilibrium_choice, Nash_equilibrium_result
>
> def calc_conditional_NEs(task_id, ban, banp):
>     info = json.load(open(f"dataset/families/{task_id}.json"))
>     pA, pB = info["row payoffs"], info["column payoffs"]
>     return get_Nash_equilibrium(pA, pB, ban, banp)[0]
>
> actual_optimal_situation_pre_task = calc_conditional_NEs(pre_task_id, restricted_situation, restricted_player)
> ```
>
> We hope these clarifications address your questions and look forward to further discussions and receiving your valuable guidance and feedback. Thank you!

---

> > ### Author Response · Authors · 2024-12-02
> > **Thank you for your review and feedback**
> >
> > Dear reviewer SyXy,
> >
> > Thank you for your review and constructive feedback! We hope that our responses have sufficiently addressed the questions and concerns you raised. We would greatly appreciate your continued support and any additional comments or suggestions you may have.
> >
> > Best,
> >
> > Paper 6932 Authors

---

### Official Review · Reviewer_cWxi · 2024-11-04

**Soundness:** 3
**Presentation:** 2
**Contribution:** 4
**Rating:** 5
**Confidence:** 4

**Summary:**

The authors create TMGBench, a game theory based benchmark for testing the strategic reasoning abilities of LLMs. They create a large number of games based on the "Robinson-Goforth topology of 2x2 matrix games" as well as utilizing synthetic data generation to build on top of said games for further game development. The games are then combined in a variety of ways, creating a complex structure for the LLMs to reason in. The authors then evaluate a selection of LLMs on the benchmark and report their results.

**Strengths:**

- Models are tested rigorously; 2,880 times for a single model in the single game tests, the complex games have a baseline of being tested 20 times, and there's testing for positional bias with the reFoToM / reSoToM prompts.
- Extensibility: this is a great way of creating a difficult-to-overfit-to benchmark, using the synthetic data generated stories as additional "games" to play.
- The metrics used (ID, BD, PAR) are comprehensive for evaluating a model's performance and good insight to how the models perform in these situations.
- The tables and figures nicely present the findings of the experiments and are mostly given good descriptions.

**Weaknesses:**

- The paper can be hard to follow at times. It would be nice to have examples of the complex games to solidify the reader's understanding. The description given for sequential games doesn't quite make sense to me, even with two introductions. And because of that, I'm not sure how well it upholds the task of "testing for strategic reasoning".
- I'm not convinced that parallel forms are actually a test of strategic reasoning either, this seems closer to measuring the model's "working memory" and being able to keep track of the different situations at a given time step. But, this may be based on a misunderstanding of what the form is describing; it's not clear to me based on the descriptions given.
- The prompt given for `Example of classic game: classic/111` gives me pause for the rest of the prompt generation. "Player A and Player B are playing a game. Either of them has two choices, namely A1, A2/B1, B2." Is this telling the model that the choices are {A1, A2} or {B1, B2}? I assume this, but that could lead to the model being confused about the task rather than being honestly judged on the difficulty of the task.

- a number of simple proofreading errors:
	- "sequential form, where LLMs are required to response multiple game tasks in a row" --> "to respond to multiple games"
	- "As explained in Section 2.2, our benchmark are perfectly suitable" --> your benchmark what?
	- "as for practical result provide by LLMs," --> results provided by
	- "which we expect robuster LLMs" --> "more robust LLMs", I'm not sure if "robuster" is a word, but if it is it's not commonly used.
		- "using CoT prompting, which is robuster"
	- "We perform 4 independent tests on each data point, covering both the classic setting and the story-based setting. Basically, we conduct 2,880 tests to generally evaluate a certain model"
		- this is weird, "Basically, we conduct 2,880 tests..." these should be combined to make flow better.
	- "We setup the test by divided it into several types" --> "by dividing it"

**Questions:**

- interesting that llama70B did worse on DA than 8B, why do you think this is?

---

> ### Author Response · Authors · 2024-11-17
> **Thanks for your careful review!**
>
> We sincerely thank you for your detailed comments and constructive feedback. Below, we address the specific concerns you raised:
>
> **Question**: The sequential/parallel form games seem not to hold the task of testing for strategic reasoning.
>
> **Response**: We sincerely apologize for not clearly articulating the significance and importance of complex games in our benchmark. It is worth further emphasizing that our motivation for designing complex form games is to highlight that the atomic games in TMGBench resemble the “primary components” of complex real-world social scenarios.
> Specifically:
> - For sequential games, real-life scenarios often require making decisions one after another to solve problems.
> - For parallel games, such as in diplomatic contexts, governments often need to simultaneously make decisions in multiple domains, including technology, military, politics, and culture.
> - For nested games, as seen in scenarios like auctions, the decisions made in earlier auctions often influence subsequent ones.
>
> Thus, the complex game forms we designed can effectively represent various complex strategic scenarios in real life, enabling a more in-depth evaluation of large language models’ strategic reasoning capabilities.
>
> ---
>
> **Question**: Potential ambiguous prompt expression.
>
> **Response**: We understand your concerns. Prior to our formal experiments, we conducted extensive prompt engineering to ensure that the prompts we currently use can reliably test the models. Additionally, the experimental results in our formal tests have validated that the prompts effectively and consistently evaluate the models.
>
> ---
>
> **Question**: Proofreading errors.
>
> **Response**: We are grateful that you meticulously pointed out parts of the text that may not be easy for readers to understand. We have revised and improved these sections in the paper accordingly based on your guidance (respectively on line 124, line 224, line 235, line 364, line 317, line 476).
>
> ---
>
> **Question**: The potential cause of Llama70B performing worse on DA than 8B.
>
> **Response**: We acknowledge your observation regarding the underperformance of the DA prompt (Llama70B performing worse than Llama8B). Here is our interpretation of this finding:
> - Positional Bias: One possible explanation is positional bias. In some cases, larger models may exhibit stronger biases towards certain choices rather than random ones, leading to suboptimal performance. We will analyze this phenomenon further in the data.
> - Emergent Abilities: The poor performance of both Llama70B and Llama8B under the DA prompt suggests that these models do not exhibit emergent abilities without CoT prompting. This highlights the limitations of current LLMs in strategic reasoning tasks, even as model size increases.
> Future Work: We plan to include additional statistical analysis to strengthen our interpretation and discuss this observation more thoroughly in the experimental results section of the revised paper.
>
> We look forward to engaging in further discussions with you and receiving your additional guidance and feedback. Thank you!

---

> > ### Comment · Reviewer_cWxi · 2024-11-26
> >
> > Thank you for addressing the comments I brought up.
> >
> > > re: sequential/parallel form games for strategic reasoning
> > - Thank you for elaborating on this point. From your comment it's not clear if the "atomic games as primary components of complex social scenarios" idea was already mentioned in the paper and I overlooked it. If it wasn't included in the paper, adding this would shift my evaluation of the paper and preferably including references that establishes this notion.
> >
> >
> > > Potential ambiguous prompt expression
> > - You mention, "we conducted extensive prompt engineering to ensure that the prompts we currently use can reliably test the models". I'd want to see the metrics you decided on that satisfied the "reliably testing the models". Currently the reader has no insight as to what was decided upon for a prompt being "satisfactory" or what the search space of prompting techniques were. It may be the case that there's little prompting research for this niche (which I honestly doubt, there's a decent amount of game theory literature that has likely looked into relevant prompting techniques), but mentioning that lack of previous research along with your guiding principles for 1. developing and 2. accepting a prompt would give the reader context on what to expect here.
> > - Note, I'm focusing on these prompts as they are the medium through which model insights become available to us. It may be the case that a model will show greatly different results across the benchmark given different prompting strategies, and being able to track this gives the reader credence to what attention was paid to this during this study.
> >
> > > proofreading
> > - One suggestion I'd make for making paper updates during rebuttals is to include the changes in red text to make easier for the reviewer to identify the exact changes; a "visual diff" of sorts. Just a small note for future reference.
> >
> > Finally, there was no mention of making the paper easier to follow (adding game examples, description of the sequential games). I would still be interested in seeing this.

---

> > > ### Author Response · Authors · 2024-11-28
> > > **Thanks for your follow-up comment!**
> > >
> > > Thank you for the follow-up comment! We address the issues as follow:
> > >
> > > ---
> > >
> > > **Advice: Adding References About the Role of Atomic Games**
> > >
> > > Thank you for your helpful advice. In response, we have added a subsection in **appendix F** to clarify the role of atomic games, along with the relevant reference, to better explain their significance in our work.
> > >
> > > ---
> > >
> > > **Question: Metrics of a "Satisfactory" Prompt**
> > >
> > > We conducted extensive testing on the prompt, focusing on three key aspects:
> > >
> > > 1. **Task Validity**: Task validity refers to how well the LLM understands the task described in the prompt and follows the format requirements to produce an error-free response. A satisfactory prompt should ensure high task validity, meaning the model can comprehend the task and generate appropriate outputs with error less frequently.
> > >
> > > 	This metric is represented by *accuracy rate (acc)*:
> > > 	$$
> > > 	acc = \frac{\sum_{i=1}^N \mathbb{I} ( parsable(r_i) ) }{N}
> > > 	$$
> > > 	where $N$ means the sampled query times, $r_i$ refers to the $i$-th response, $parsable(\cdot)$ is the function to check if the response have correct format. The larger *acc* means higher task validity.
> > >
> > > 2. **Response Consistency**: Response consistency measures the stability of the LLM's responses when given the same prompt multiple times. Our goal in prompt engineering is to minimize **aleatoric uncertainty**, which refers to the inherent randomness or noise in the prompt itself. While consistency is important, it's crucial to note that **epistemic uncertainty**—which stems from the model's limitations—may still remain, even with a fixed prompt. This highlights the model’s true capabilities and its potential limitations in handling certain tasks.
> > >
> > > 	This metric is represented by *gini index (gini)*:
> > > 	$$
> > > 	gini = 1 - \sum_{i=1}^T p_i^2
> > > 	$$
> > > 	where $T$ is the number of types of answer (derived from error-free response) in total tries, $p_i$ indicates the proportion of the $i$-th type of answer to the total tries. The larger *gini* means higher response consistency.
> > >
> > > 3. **Paraphrase Similarity**: Paraphrase similarity evaluates how well the LLM maintains the quality and correctness of its responses when the prompt is paraphrased  with similar meaning. A satisfactory prompt should ensure that the model generates consistent responses, regardless of how the task is phrased, demonstrating its robustness to variations in input phrasing.
> > >
> > > 	This metric is represented by *average deviation (dev)*:
> > > 	$$
> > > 	dev = \frac{\sum_{i=1}^M sim(R_P, R_{P_i})}{M}
> > > 	$$
> > > 	where $M$ is the number of paraphrased prompt of target prompt $P$, $P_i$ indicates the the prompt of the $i$-th paraphrase. $sim(\cdot, \cdot)$ is the function computing the similarity of two response set (using $R$ to represent response set). The smaller *dev* means higher paraphrase similarity.
> > >
> > > Regarding the candidate prompt, we first manually constructed some and then iteratively sampled alternative prompts by paraphrasing and rephrasing them with different LLMs.
> > >
> > > ---
> > >
> > > **Question: No Further Demonstration of Complex Form Games**
> > >
> > > We apologize for the inconvenience caused by the lack of concrete examples for the complex form games, which might have made the paper hard to follow. To clarify, we have added a new figure in a subsection of **appendix F** to better illustrate the different complex forms and their application in our framework. We hope this addition will provide a clearer understanding of how these games work.
> > >
> > > ---
> > >
> > > We look forward to engaging in further discussions with you and receiving your additional guidance and feedback. Thank you!

---

> > > > ### Author Response · Authors · 2024-12-02
> > > > **Thank you for your review and feedback**
> > > >
> > > > Dear reviewer cWxi,
> > > >
> > > > Thank you for your review and constructive feedback! We hope that our responses and revisions have addressed the concerns you raised. Upon your advice, we have already made our latest revision error-free and include more figures in the appendix sections to make the paper easier to follow. Please feel free to share any additional comments or suggestions. We greatly appreciate your thorough review and continued support.
> > > >
> > > > Best,
> > > >
> > > > Paper 6932 Authors

---

> > > > > ### Author Response · Authors · 2024-12-03
> > > > > **Look forward to your new feedback**
> > > > >
> > > > > Dear reviewer cWxi,
> > > > >
> > > > > We are very concerned whether our response has addressed your concerns and look forward to your new feedback.
> > > > >
> > > > > Best,
> > > > >
> > > > > Paper 6932 Authors

---

> > > > ### Comment · Reviewer_cWxi · 2024-12-03
> > > >
> > > > Thank you for addressing my comments and questions.
> > > >
> > > > I have updated my score from:
> > > > - Presentation: 1 --> 2
> > > > - Contribution: 3 --> 4
> > > >
> > > > Stating the soundness on game composition makes this benchmark more robust. Also, including the examples helps the paper make more sense.
> > > >
> > > > You do a great job of explaining your prompt metrics, data should be collected on this and included (along with these descriptions) in the paper itself.

---

> > > > > ### Author Response · Authors · 2024-12-03
> > > > > **Seek for improvement of overall rating**
> > > > >
> > > > > Dear reviewer cWxi,
> > > > >
> > > > > Thank you very much for your response and for your positive feedback on our work! We would appreciate any further suggestions you might have to help improve the overall rating. If no additional concerns remain, we kindly request that you consider updating the rating or recommendation, as we believe the previous issues have been thoroughly addressed.
> > > > >
> > > > > Best,
> > > > >
> > > > > Paper 6932 Authors

---

### Official Review · Reviewer_AuNq · 2024-11-04

**Soundness:** 2
**Presentation:** 3
**Contribution:** 2
**Rating:** 5
**Confidence:** 3

**Summary:**

The paper introduces TMGBENCH, a benchmark for systematically evaluating the strategic reasoning abilities of LLMs. By evaluating some LLMs on TMGBENCH, the paper identifies several flaws in LLMs’ performance, such as low accuracy rates and unstable inconsistency.

**Strengths:**

- The paper is well written and well organized.
- The games included in TMGBENCH are comphrehensive.

**Weaknesses:**

- I am not fully convinced there exists the need for a benchmark fo evaluating strategic reasoning abilities of LLMs. In fact, there lacks an universal definition of the ability of strategic reasoning. In other words, what are the fundemental differences between tasks that require strategic reasoning and tasks that do not?


- If there is a clear definition of strategic reasoning, I would expect a more systematic study of existing LLMs on strategic reasoning. Why some LLMs perform better than others in terms of strategic reasoning? What are the influencing factors of LLMs? Data, Architecture, Model Size, training objectives?

**Questions:**

Regarding weakness 1:

- Do you have a clear definition of tasks that require strategic reasoning, as used in this paper?

- Could you explain more on how TMGBENCH addresses gaps in existing benchmarks for evaluating LLM reasoning capabilities?

- What are the fundemental differences between tasks that require strategic reasoning and tasks that do not, perhaps with concrete examples?

Regarding weakness 2:

- Could you conduct an analysis of how different LLM characteristics (e.g., model size, architecture, training data, or objectives) correlate with performance on TMGBENCH? and why.

---

> ### Author Response · Authors · 2024-11-21
> **Thanks for your careful review! (1/2)**
>
> First, we thank you for giving kind advice and insightful opinions, and we will address your questions one by one.
>
> Of Weakness 1:
>
> **Question:** Lack of a clear definition for strategic reasoning.
>
> **Response:** We fully understand your concern about the lack of a clear definition of strategic reasoning and would like to provide further clarification. In the introduction of our paper, we referenced a survey paper titled "A Survey of Strategic Reasoning with Large Language Models", which defines strategic reasoning and discusses the key characteristics of tasks that require it. We used the concept derived from that, and we have already added more detailed explanations in the appendix to make it clearer for readers. Please refer to Appendix A of the latest revised paper. Here is a short explanation:
> 1. *Definition of Strategic Reasoning*: Strategic reasoning can be defined as the ability to make decisions based not only on one’s own actions but also on predicting and responding to the actions of others, especially in environments where outcomes depend on the interdependence of agents’ decisions. This definition distinguishes strategic reasoning from other forms of reasoning, such as common-sense reasoning or logical reasoning.
>
> 2. *Characteristics of Strategic Reasoning*: The core feature of strategic reasoning is its reliance on anticipating and responding to the behavior of other participants.
>
> 3. *Necessity and Applications of Strategic Reasoning*: Strategic reasoning is vital in various fields, including decision-making intelligence and social intelligence. For example, strategic reasoning is required in games like poker and chess, where players must predict and counter their opponents’ moves.
>
> ---
>
> **Question:** Differences Between Our Benchmark and Existing Ones.
>
> **Response:**
>
> 1. *Game Coverage*: TMGBench focuses on a specific but symmetric subset of 2x2 matrix games. Rather than incorporating a wide range of game types, we diversify the configurations and descriptions within this subset. While these games may share a common form representation, slight changes in numerical parameters or contextual framing can result in significantly different challenges for LLMs. Our experiments demonstrate that LLMs often become confused and perform poorly when only a few numbers are altered, highlighting their generalization issues in strategic reasoning.
>
> 2. *Data Volume*: Each game in TMGBench contains several data points, ensuring a more comprehensive evaluation. This diversity in examples helps create a more robust and stable evaluation.
>
> 3. *Scenario Diversity*: TMGBench introduces a wide range of scenarios within its game structures, offering diverse contexts in both numeric setups and story-based reframing. For example, existing benchmarks often focus solely on abstract payoff matrices, whereas TMGBench incorporates narrative elements to test how well LLMs adapt to different descriptions of the same strategic challenges. This diversity reflects real-world complexities, where decisions are rarely presented in isolated, numeric forms but instead in nuanced, contextual settings.
>
> 4. *Game Extensibility*: TMGBench enables researchers to expand the benchmark by introducing additional forms of game compositions or more complex scenarios. For example, nested games can simulate hierarchical decision-making scenarios, like auctions or negotiations. This extensibility allows TMGBench to adapt to future developments in strategic reasoning research, offering long-term utility for evaluating increasingly sophisticated LLMs.
>
> ---
>
> **Question:** Differences Between Strategic Reasoning and Other Types of Reasoning.
>
> **Response:**
> Strategic reasoning differs fundamentally from other types of reasoning, such as common-sense reasoning, in that it involves considering the actions of other participants and predicting their behavior. For example, in common-sense reasoning, the focus is on making inferences based on factual knowledge, while in strategic reasoning, the focus is on understanding and anticipating the intentions and actions of other participants.

---

> > ### Author Response · Authors · 2024-11-21
> > **Thanks for your careful review! (2/2)**
> >
> > Of Weakness 2:
> >
> > We appreciate your suggestion for a more systematic analysis of how different LLM characteristics influence performance on TMGBench. We agree that such an analysis is crucial for understanding the factors that contribute to strategic reasoning abilities in LLMs. To address this concern, we performed additional experiments. Below are the results:
> >
> > | Model            | PAR(↑)             | ID(↓)              | BD(↓)             | Difference                  |
> > | ---------------- | ------------------ | ------------------ | ----------------- | --------------------------- |
> > | Qwen2-7B         | 21.30              | 22.72              | 39.71             | (baseline)                  |
> > | Qwen2-Math-7B    | **63.17 (+41.87)** | **7.87 (-14.85)**  | **14.47(-25.24)** | tuning dataset              |
> > | Qwen2-72B        | 46.21              | 19.94              | 29.29             | model size                  |
> > | Qwen2-Math-72B   | **57.81 (+36.51)** | **11.07 (-11.65)** | **19.64(-20.07)** | tuning dataset & model size |
> > | Qwen2.5-7B       | 48.26              | 11.51              | 18.43             | generation                  |
> > | Qwen2.5-Coder-7B | 44.62              | 13.27              | 26.48             | tuning dataset              |
> > | WizardLM-7B      | 21.24              | 28.84              | 57.52             | architecture                |
> >
> > From the table, we observe that
> > 1. *Effect of Dataset Tuning*: Math-tuned datasets significantly enhance performance in strategic reasoning tasks, while code-tuned datasets show limited improvements.
> > 2. *Effect of Model Size*: Larger models (e.g., 72B) generally perform better than smaller ones (e.g., 7B). However, the marginal benefit of size increases appears to be smaller compared to the benefit from dataset tuning.
> > 3. *Effect of Training Process and Objectives*: Models trained with different objectives (e.g., Qwen2-7B vs. WizardLM-7B) exhibit notable performance differences, highlighting the impact of pretraining strategies.
> >
> > We hope these additional experiments provide insights for addressing your concern, and we plan to conduct more comprehensive experiments in our future work.
> >
> > We look forward to engaging in further discussions with you and receiving your additional guidance and feedback. Thank you!

---

> > > ### Comment · Reviewer_AuNq · 2024-11-26
> > >
> > > Thanks very much for the rebuttal, which alleviates some of my concerns. Yet, I still find the contribution of the paper not strong enough for the following two reasons. Hence, I will maintain the initial rating.
> > > - In the rebuttal, the authors said "Strategic reasoning differs fundamentally from other types of reasoning, such as common-sense reasoning, in that it involves considering the actions of other participants and predicting their behavior."  The authors mentioned poker and chess as examples. Yet, anticipating other players actions is not a necessity for playing well in poker. In fact, existing powerful AIs in poker don't predict other players actions, such as DeepStack, Libratus, Pluribus. For chess, it is a perfect information game, and there is no need to predict other players actions. To summarise, I am not fully convinced of the significance of the new benchmark presented in this paper.
> > >
> > > - The experimental analysis in the rebuttal about how different LLM characteristics (e.g., model size, architecture, training data, or objectives) correlate with performance on TMGBENCH makes a good starting point but still looks incomplete. I would expect a more thorough investigation and ablations.

---

> > > > ### Author Response · Authors · 2024-11-28
> > > > **Thanks for your follow-up comment!**
> > > >
> > > > Thanks for your follow-up comment! We greatly appreciate the time and constructive comments you have provided.
> > > >
> > > > ---
> > > >
> > > > **Concern 1: Does AI need strategic reasoning in games like poker or chess?**
> > > >
> > > > We understand your concern that state-of-the-art models in games like poker and chess may not need to explicitly predict every move of other players. Indeed, reinforcement learning (RL) models, such as those used in DeepStack, Libratus, and Pluribus, optimize their strategies based on probabilistic reasoning and game-theory principles, without explicitly predicting each opponent’s move.
> > > >
> > > > However, large language models (LLMs) differ from these RL-based models in several ways. LLMs do not perform exhaustive search within the game tree, nor do they have a precise reward model. Instead, they generate responses based on patterns learned from large datasets, making their decision-making more analogous to human players. In games like chess or poker, an LLM-driven player does not calculate all possible outcomes exhaustively but predicts moves based on contextual understanding—much like humans who reason strategically, even without full information.
> > > >
> > > > In a previous study [1] on LLMs and strategic games like UNO, it was found that while RL models generally excel at strategy optimization, models like GPT-4, when enhanced with reflection modules, can outperform RL models in certain strategic contexts. However, most LLMs still lag behind RL models in terms of overall performance. This underscores both the current limitations and the untapped potential of LLMs in complex game scenarios.
> > > >
> > > > Additionally, we would like to clarify that the application of *Theory of Mind (ToM)* plays a critical role in these scenarios, whether for human players or LLM-based agents. ToM enables an agent to consider the mental states of other participants, which is crucial for strategic reasoning. In another study [2], LLM agents were elicited to utilize ToM capabilities for playing Guandan, a variant of poker, and it was shown that incorporating ToM consistently improves their performance against opposing agents.
> > > >
> > > > As for chess, while it is a perfect-information game, the state space is vast, and LLMs, like human players, rely on heuristic reasoning rather than exhaustive search. Due to this, LLMs may not be able to explore the entire state space, but they can still reason strategically by leveraging learned patterns.
> > > >
> > > > ---
> > > >
> > > > **Concern 2: Seemingly Incomplete Investigation On How Different Factors Affect Models Performance**
> > > >
> > > > We acknowledge that the initial findings presented are a starting point and that a more detailed exploration of how different LLM characteristics influence performance on TMGBench would be valuable. In our current study, we focused on a limited set of factors due to some constraints (time, cost, etc.), but we are planning to expand this analysis in future work, incorporating more comprehensive ablation studies and further investigations into model architecture, training data, and model size. We hope to provide a more thorough analysis in future iterations of this research, which will enhance the robustness of our findings and contribute to the overall understanding of LLMs' strategic reasoning abilities.
> > > >
> > > > ---
> > > >
> > > > We hope this clarifies our approach and the future directions we plan to pursue. Once again, thank you for your feedback, and we value your continued engagement with our work!
> > > >
> > > > [1] Qin, Zhanyue, et al. "UNO Arena for Evaluating Sequential Decision-Making Capability of Large Language Models." *Proceedings of the 2024 Conference on Empirical Methods in Natural Language Processing*. 2024.
> > > >
> > > > [2] Yim, Yauwai, et al. "Evaluating and enhancing llms agent based on theory of mind in guandan: A multi-player cooperative game under imperfect information." *arXiv preprint arXiv:2408.02559* (2024).

---

> > > > > ### Author Response · Authors · 2024-12-02
> > > > > **Thank you for your review and feedback**
> > > > >
> > > > > Dear reviewer AuNq,
> > > > >
> > > > > Thank you for your review and constructive feedback! We hope that our responses and revisions have addressed the concerns you raised. We have carefully explained our viewpoint in comments and incorporated additional sections to make our standpoint more clearly. Please feel free to share any additional comments or suggestions. We greatly appreciate your thorough review and continued support.
> > > > >
> > > > > Best,
> > > > >
> > > > > Paper 6932 Authors

---

> > > > > > ### Author Response · Authors · 2024-12-03
> > > > > > **Look forward to your new feedback**
> > > > > >
> > > > > > Dear reviewer AuNq,
> > > > > >
> > > > > > We are very concerned whether our response has addressed your concerns and look forward to your new feedback.
> > > > > >
> > > > > > Best,
> > > > > >
> > > > > > Paper 6932 Authors

---

> > > > > > > ### Author Response · Authors · 2024-12-03
> > > > > > > **seek for latest feedback**
> > > > > > >
> > > > > > > Dear reviewer AuNq,
> > > > > > >
> > > > > > > Given the rebuttal deadline, we kindly request your latest feedback at your earliest convenience. Thank you for your understanding and prompt attention to this matter.
> > > > > > >
> > > > > > > Best,
> > > > > > >
> > > > > > > Paper 6932 Authors

---

### Meta-Review · Area_Chair_o8cs · 2024-12-23

**Metareview:**

The paper introduces an LLM benchmark based on game theory games (prisoners dilemma and its ilk, not games that people actually play). Reviewers are lukewarm about the paper, partly because of a perceived dearth of novelty and justification, which I consider not to be reason enough to reject the paper. More importantly in this case, there seems to be quite a few outstanding questions about many issues, including potential data leakage for some games, the differentiation between this and other similar benchmarks, and the clarity of the prompts. The authors attempted to address the concerns of the reviewers, but not always in a convincing way.

**Additional Comments On Reviewer Discussion:**

The authors attempted to address the concerns of the reviewers, but not always in a convincing way.

---

### Decision · Program_Chairs · 2025-01-22

Reject